# UltraEdit: Instruction-based Fine-Grained Image Editing at Scale

**Haozhe Zhao**[1,3*], **Xiaojian Ma**[2*], **Liang Chen**[1,5], **Shuzheng Si**[4], **Rujie Wu**[5], **Kaikai An**[1,3],
**Peiyu Yu**[6], **Minjia Zhang** [7], **Qing Li**[2✉], **Baobao Chang**[1✉]

[1]National Key Laboratory for Multimedia Information Processing, Peking University
[2]State Key Laboratory of General Artificial Intelligence, BIGAI
[3]School of Software and Microelectronics, Peking University
[4]Department of Computer Science and Technology, Tsinghua University
[5]School of Computer Science, Peking University, China,   [6]UCLA, [7]UIUC
*Equal contribution   ✉ Corresponding author
mimazhe55360@gmail.com,{maxiaojian,liqing}@bigai.ai,chbb@pku.edu.cn
ultra-editing.github.io

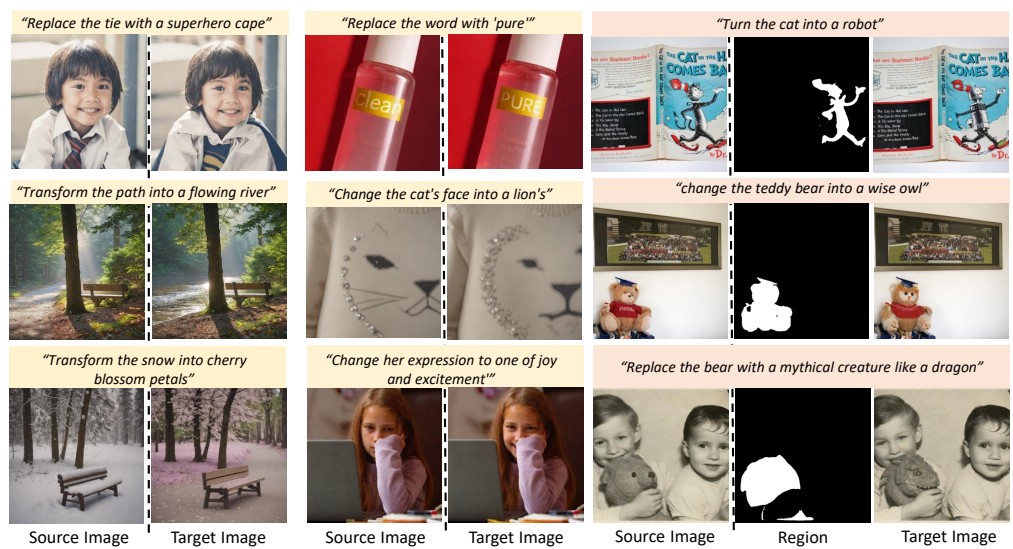

Figure 1: **Examples of ULTRAEDIT**. Free-form (left) and region-based (right) image editing.

## Abstract

This paper presents ULTRAEDIT, a large-scale (~4M editing samples), automatically generated dataset for instruction-based image editing. Our key idea is to address the drawbacks in existing image editing datasets like InstructPix2Pix [10] and MagicBrush [71], and provide a *systematic* approach to producing massive and high-quality image editing samples. ULTRAEDIT offers several distinct advantages: 1) It features a broader range of editing instructions by leveraging the creativity of large language models (LLMs) alongside in-context editing examples from human raters; 2) Its data sources are based on real images, including photographs and artworks, which provide greater diversity and reduced bias compared to datasets solely generated by text-to-image models; 3) It also supports region-based editing, enhanced by high-quality, automatically produced region annotations. Our experiments show that canonical diffusion-based editing baselines trained on ULTRAEDIT set new records on MagicBrush and Emu-Edit benchmarks. Our analysis further confirms the crucial role of real image anchors and region-based editing data. The dataset, code, and models are available in github.com/pkunlp-icler/UltraEdit.

38th Conference on Neural Information Processing Systems (NeurIPS 2024) Track on Datasets and Benchmarks.

# 1 Introduction

We present a new dataset, ULTRAEDIT, for instruction-based image editing at scale (see Figure 1 for examples, more in Appendix B.2). Compared to several prior works on the same front [10, 60, 71], we venture into tackling some of their drawbacks, which prevent the model from robustly interpreting and executing instructions. We summarize our observations on their downsides below:

- **Limited instruction diversity.** Prior works have employed two strategies to produce editing instructions: 1) To use human raters, *e.g.*, MagicBrush [71], which offer human-aligned and diverse instructions, but could be challenging to scale to a massive amount; 2) LLM-assisted generation, *e.g.*, InstructPix2Pix [10], which is scalable but the instruction variety could be limited (bounded by the relevant capabilities of LLMs); thus making it hard to generalize to novel instructions.

- **Implicit biases in images.** Due to the challenges of obtaining massive instruction-based image editing data, previous research has widely adopted text-to-image (T2I) models [49, 48, 50, 51, 56, 24] to produce *both* the source images and target (edited) images. However, many T2I models could suffer from implicit biases [32, 7, 43, 19, 9, 15, 66], *i.e.* they produce images subjected to certain domains or characteristics more than others. Therefore, the resulting image editing dataset could be quite unbalanced. When this dataset is used for training, the trained model's performance could suffer in image domains that T2I models do not cover well. For example, if a T2I model tends to generate cartoon-style images, models trained on the resulting data may perform poorly when editing images depicting natural scenes.

- **Missing of region-based editing.** Most of the existing datasets only consider free-form editing, *i.e.*, the model is given a source image and an instruction only, while many of the actual image editing scenarios also involve a region where the editing is expected to happen (a "mask"). We will later show that such additional region guidance could significantly boost the editing performance. Therefore, missing such region data could hinder the quality of models trained on the dataset.

To address these limitations, we propose a *systematic* approach to curate massive and high-quality image editing data automatically. An overview of our method can be found in Figure 2. Specifically, we begin by using LLMs along with in-context human-written instruction examples to ensure *diverse* editing instructions. Next, we use prompt-to-prompt (P2P) [24] control with off-the-shelf T2I diffusion models to produce source and target (edited) images from captions and editing instructions. However, to *avoid the biases* within the T2I models, we collect high-quality image-caption pairs from diverse real image datasets like COCO [37] and use these images as *anchors* to guide the T2I models when producing source images given the corresponding caption. Additionally, we employ an automatic region generation approach to produce editing regions from the instruction and utilize such region annotations in a modified inpainting diffusion pipeline to produce *region-based editing* samples. As a result, the final dataset, ULTRAEDIT, comprises ~4M editing samples with ~750K unique instructions across 9+ editing types. To the best of our knowledge, ULTRAEDIT is the largest instruction-based image editing dataset to be released to the public. With ULTRAEDIT, canonical diffusion-based editing baselines enjoy new records on challenging MagicBrush and Emu-Edit benchmarks, respectively. Our analysis further confirms the crucial role of real image anchors and region-based editing data. To sum up, our contributions can be summarized as follows:

- We propose a novel automatic pipeline to generate image editing data, mitigating the issues of existing datasets: instruction diversity, image biases, and missing region-based editing data.
- With our pipeline, we curate ULTRAEDIT, a large-scale and high-quality image editing dataset with diverse instructions, real images as anchors, and additional editing region annotations.
- We conduct extensive studies on how canonical diffusion-based image editing model can benefit from ULTRAEDIT and provide insights and analysis on the key design principles.

# 2 The ULTRAEDIT Dataset

## 2.1 Dataset Formation

In ULTRAEDIT, we consider two editing settings: 1) free-form editing, where the editing could happen at any area of the input image; 2) region-based editing, where the editing is expected to be about a certain region of the image, allowing more *fine-grained editing*. In all, our dataset can be formulated as a set of $\langle I_s, I_t, T_e, I_m, T_s, T_t \rangle$, where $I_s$ and $I_t$ denote source and target (edited) image, respectively, $T_e$ is the editing instruction, $I_m$ denotes the additional editing region (mask). The model

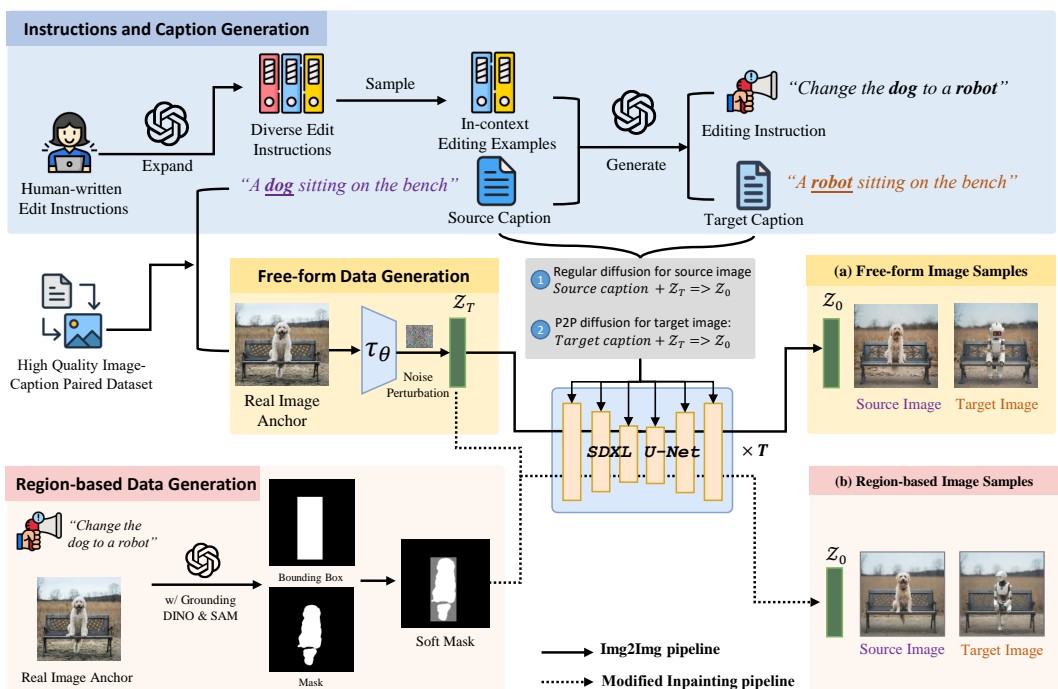

Figure 2: **The construction of ULTRAEDIT**. (Upper) We use LLM with in-context examples to produce editing instructions and target captions given the collected image captions; (Middle) For free-form editing, we use the collected images as anchors, and invoke regular diffusion followed by prompt-to-prompt (P2P) [24] control to produce source and target images; (Bottom) For region-based editing, we first produce an editing region based on the instruction, then invoke a modified inpainting diffusion pipeline [44, 5] to produce the images.

is expected to map editing input $I_s$, $T_e$ and mask $I_m$ to $I_t$. We also need the captions of source and target image $T_s$, $T_t$ for image generation and evaluation purposes. Below, we detail how to collect ULTRAEDIT. An overview of our pipeline can be found in Figure 2.

**Stage I: Instruction Generation.** As we mentioned in Section 1, our key idea to address the issue of limited instruction diversity in existing datasets is to obtain high-quality editing instructions by combining LLM creativity and human raters. Specifically, we begin by manually creating hundreds of editing instructions with our human raters. The raters are first given images and captions from the COCO dataset [37] as context and asked to write appropriate editing instructions for these scenarios. We then invoke LLM to expand these human-written instructions into more diverse examples for editing in terms of semantics and editing tasks. We end up with ~10K instruction examples after the expansion. Details on the prompt and examples can be found in Appendix A.2.

**Stage II: Free-form Editing Samples.** We follow the main idea of prior work and invoke off-the-shelf T2I models to generate source and target images $I_s$, $I_t$ in editing samples. However, instead of synthesizing all images using T2I models as in prior works [10, 60], ULTRAEDIT adopts real images as anchors to mitigate the biases within these T2I models. Specifically, we first collect ~1.6M high-quality and diverse image-caption paired data from real image datasets like COCO [37], NoCaps [3], *etc.* (full details on the mix can be found in Appendix A.1). For each image-caption pair $\langle I^\star, T_s \rangle$, we sample in-context examples from the instruction pool obtained in Stage I and prompt LLM to produce the editing instruction $T_e$ and the resulting target caption $T_t$ after the editing. We then invoke a regular Img2Img diffusion pipeline with the noise-perturbed latent embedding of $I^\star$ as $z_T$ (similar to SDEdit [41]) and the source caption $T_s$ as a condition; therefore, the produced source image $I_s$ will resemble the anchor $I^\star$, *i.e.*, we use diverse real images to guide the generation of T2I models, mitigating their biases. After producing $I_s$, we invoke prompt-to-prompt (P2P) control [24] with the target caption $I_t$ to produce the target image $I_t$ on the same $z_T$ from real image anchor $I^\star$. Due to space limits, we refer the readers to the P2P paper [24] for details. We utilize SDXL-Turbo [56] as our diffusion backbone, which allows for high-quality generation with just 2-4 diffusion steps, maintaining a generation quality comparable to SDXL [46].

Table 1: **Comparison of different image editing datasets**. Both EditBench and MagicBrush are manually annotated but are limited in size. InstructPix2Pix and HQ-Edit are large datasets automatically generated using T2I models like Stable Diffusion [51] and DALL-E [50], though they present notable biases from the generative models leading to failure cases. ULTRAEDIT offers large-scale samples with rich editing tasks and fewer biases.

| Datasets | Real Image Based | Automatic Generated | Editing Region | #Edits | #Editing Types | Source Example | Instruction | Target Example |
|---|---|---|---|---|---|---|---|---|
| EditBench [68] | ✓ | ✗ | ✓ | 240 | 1 |  | *an amber vase with a narrow lip and a wide base* |  |
| MagicBrush [71] | ✓ | ✗ | ✓ | 10,388 | 5 |  | *replace the dove with an owl.* |  |
| HQ-Edit [26] | ✗ | ✓ | ✗ | 197,350 | 6 |  | remove the chisel. |  |
| InstructPix2Pix [10] | ✗ | ✓ | ✗ | 313,010 | 4 |  | *make it a stone bridge* |  |
| **ULTRAEDIT** | ✓ | ✓ | ✓ | **4,108,262** | **9+** |  | *Change the hat into a crown.* |  |

**Stage III: Region-based Editing Samples.** Upon the free-form editing samples, we create additional region-based editing data using an automatic editing region extraction method. Given an image-instruction pair $\langle I^\star, T_e \rangle$, we first detect all objects within $I^\star$ using recognize-anything [74] and prompt LLM with the object list, soruce caption $T_s$, target caption $T_t$ and editing instruction $T_e$ to identify the object to be edited. Then we employ GroundingDINO [38] and SAM [31] to obtain bounding boxes and fine-grained mask of this target object. For edits involving transformation over the entire image (*e.g.*, "turn this into an oil paint"), we mark the whole image as the editing region. We save an expanded version of the original mask (which becomes a contour) as the editing region $I_m$. Finally, the bounding box and fine-grained mask will be fused into a soft mask (see Figure 2) to help smooth the transition between inpainting area and the rest of the image. While producing the source image $I_s$ is the same as in Stage II, we adopt a modified inpainting pipeline to produce the target image $I_t$ to take the editing region $I_m$ into consideration:

$$z_{t-1} = \begin{cases} (1 - M) * z_T + M * DM(z_t) & \text{if } t \bmod 2 == 0 \\ DM(z_t) & \text{otherwise} \end{cases} \quad (1)$$

where $M$ is $I_m$ in the size of latent space, $DM(\cdot)$ denotes the diffusion model. In a nutshell, we alternate between regular diffusion and inpainting only within the mask region to guide the generation within the given region while avoiding edge artifacts, as illustrated in Figure 16. This pipeline is compatible with P2P control and the SDXL-Turbo backbone and it takes 3-7 diffusion steps to produce a target image. Further details can be found in Appendix A.3.

**Misc.** To ensure high-quality image generation, for each editing sample, we run the diffusion pipeline 100 times and filter out the deficient generations using a mixture of automatic metrics following the prior practices [10, 20] (detailed in Section 2.3). Thanks to the efficient SDXL-Turbo diffusion backbone and our implementation, our pipeline is ~100 times faster than prior work like [10].

## 2.2 Characteristics and Statistics

Our dataset contains a total of 4,108,262 instruction-based image editing data (757,879 unique edits), where free-form image editing (without region annotation $I_m$) consists of 4,000,083 instances and region-based editing includes 108,179 samples. To the best of our knowledge, this is the largest dataset to be released to the public (a comparison to other datasets can be found in Table 1). As illustrated in Table 3, ULTRAEDIT encompasses over 9 distinct editing instruction types, detailed in Table 2. The distribution of image editing instances across these types can be found in Appendix B.

## 2.3 Quality Assessment

To ensure the quality of our dataset, we employ several automatic metrics to filter out substandard images during generation following prior practices [20, 10]. First, we utilize DINOv2 similarity, CLIP

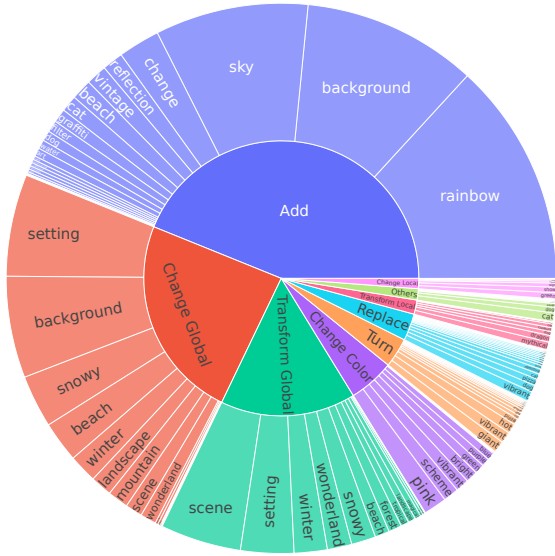

Figure 3: Distribution of edit types and keywords in the instructions of ULTRAEDIT. The inner ring illustrates the various types of edit instructions, while the outer ring presents the frequency of instruction keywords. This visualization highlights the rich diversity found within our instructions.

Table 2: Editing Instruction Types in ULTRAEDIT.

| Type | Description |
|---|---|
| **Add** | Inserting a new object or texture at a specific location in the image. |
| **Change Global** | Modifying the entire image to achieve a clear and noticeable effect. |
| **Change Local** | Altering a specific object or texture, affecting only a portion of the image. |
| **Change Color** | Adjusting the color within the image. |
| **Transform Global** | Smoothly transforming images into a different setting, scene, or style. |
| **Transform Local** | Modifying part of image features while preserving its overall structure. |
| **Replace** | Substituting existing objects in the image with those specified in the instructions. |
| **Turn** | Implicitly changing objects, background, or texture, often without a specific target. |
| **Others** | Miscellaneous editing types such as text edits and altering quantities. |

Table 3: Quantitative evaluation for ULTRAEDIT.

| Metric | Free-form. | Region-based. |
|---|---|---|
| CLIPimg | 0.8427 | 0.8813 |
| SSIM | 0.6401 | 0.7413 |
| DINOv2 | 0.7231 | 0.7688 |
| CLIPin | 0.2834 | 0.2848 |
| CLIPout | 0.3049 | 0.2848 |
| CLIPdir | 0.2950 | 0.3052 |

image similarity, and SSIM between source and target (edited) images $I_s$ and $I_t$ to guarantee that semantic similarity and pixel-level coherence can be maintained, which is crucial for image editing. Second, to ensure the generated images accurately reflect the editing instructions, we assess the alignment between the images $I_s$, $I_t$, and their corresponding captions $T_s$, $T_t$ using CLIP similarity. Finally, to verify the dataset's instruction-following capability, we employ CLIP Directional Similarity [71], which measures the alignment between changes in images and changes in corresponding captions. We provide these scores on ULTRAEDIT in Table 3. Our dataset maintains a high standard of image quality and instruction alignment across both free-form and region-based editing data. Notably, region-based data are significantly better in terms of SSIM, confirming the benefits brought by region-based guidance (via our modified inpainting pipeline). Further quality assessments can be found in Appendix B and D.

## 3 Experiments

Our experiments evaluate the quality of ULTRAEDIT in following user instructions faithfully while preserving the visual fidelity of the source image. First, we evaluate the performance of canonical diffusion-based editing models trained on our dataset across different instruction-based image editing benchmarks. Second, we dive into insights and analysis on the design principles of our dataset.

### 3.1 Setup

**Settings.** We follow the settings of Emu Edit [60] to train a diffusion model using various scales and types of data from our dataset, then evaluate the trained model across multiple benchmarks to assess the effectiveness of our dataset in advancing its image editing capabilities. For a fair comparison, we adopt the editing diffusion model introduced in InstructPix2Pix [10], which uses Stable Diffusion v1.5 [53] as the backbone diffusion model. We also employ training data volumes identical to that used in [10]. To support region-based editing, we augment the model's U-Net to take in extra channels for region (mask) input. More details can be found in Appendix A.4.

**Baselines.** We set the following models as baselines, categorized into instruction-guided image editing methods (the same setting as ours) and global description-guided methods, the latter requires descriptions of the target image to perform editing. The instruction-guided image editing methods include: InstructPix2Pix [10], HIVE [73], MagicBrush [71], and Emu Edit [60]. The global description-guided image editing methods include: Null Text Inversion [42], SD-SDEdit [41], GLIDE [44], and Blended Diffusion [5]. Notably, GLIDE and Blended Diffusion require a **region mask** for editing.

Table 4: **Results on the MagicBrush test set**. We include both single-turn and multi-turn settings. We evaluate the models trained on ULTRAEDIT without region-based editing data and full data ("Ours"). Models trained with full data are either evaluated with or without editing region as input.

| Settings | Methods | L1↓ | L2↓ | CLIP-I↑ | DINO↑ |
|---|---|---|---|---|---|
| | *Global Description-guided* | | | | |
| | SD-SDEdit | 0.1014 | 0.0278 | 0.8526 | 0.7726 |
| | Null Text Inversion | 0.0749 | 0.0197 | 0.8827 | 0.8206 |
| | GLIDE | 3.4973 | 115.8347 | 0.9487 | 0.9206 |
| | Blended Diffusion | 3.5631 | 119.2813 | 0.9291 | 0.8644 |
| **Single-turn** | *Instruction-guided* | | | | |
| | HIVE | 0.1092 | 0.0380 | 0.8519 | 0.7500 |
| | InstructPix2Pix (IP2P) | 0.1141 | 0.0371 | 0.8512 | 0.7437 |
| | IP2P w/ MagicBrush | 0.0625 | 0.0203 | **0.9332** | **0.8987** |
| | Ours, trained w/o region data | 0.0689 | 0.0201 | 0.8986 | 0.8477 |
| | Ours, eval w/o region | 0.0614 | 0.0181 | 0.9197 | 0.8804 |
| | Ours, eval w/ region | **0.0575** | 0.0172 | 0.9307 | 0.8982 |
| | *Global Description-guided* | | | | |
| | SD-SDEdit | 0.1616 | 0.0602 | 0.7933 | 0.6212 |
| | Null Text Inversion | 0.1057 | 0.0335 | 0.8468 | 0.7529 |
| | GLIDE | 11.7487 | 1079.5997 | 0.9094 | 0.8494 |
| | Blended Diffusion | 14.5439 | 1510.2271 | 0.8782 | 0.7690 |
| **Multi-turn** | *Instruction-guided* | | | | |
| | HIVE | 0.1521 | 0.0557 | 0.8004 | 0.6463 |
| | InstructPix2Pix (IP2P) | 0.1345 | 0.0460 | 0.8304 | 0.7018 |
| | IP2P w/ MagicBrush | 0.0964 | 0.0353 | 0.8924 | 0.8273 |
| | Ours, trained w/o region data | 0.0883 | 0.0276 | 0.8685 | 0.7922 |
| | Ours, eval w/o region | 0.0780 | 0.0246 | 0.8954 | 0.8322 |
| | Ours, eval w/ region | **0.0745** | **0.0236** | **0.9045** | **0.8505** |

**Benchmark and Metrics.** We evaluate the model trained on our dataset across two popular benchmarks: MagicBrush [71] and Emu Edit Test [60]. MaigicBrush benchmark evaluates the model by comparing the edited images with **ground truth images** and corresponding captions across different metrics. Following the MagicBrush [71], we chose the L1 distance, L2 distance, CLIP image similarity, and DINO similarity as metrics. Emu Edit Test benchmark compares the edited images with the **source image** and target captions for evaluation. Consistent with the Emu Edit [60], we use L1 distance, CLIP image similarity, DINO similarity, CLIP text-image similarity, and CLIP text-image direction similarity as metrics. These metrics generally measure how the edited image both preserves the original's style and content and reflects modifications according to instructions. Details of the benchmark and metrics can be found in Appendix C.

## 3.2 Main Result I: General Image Editing on MagicBrush

We present the results on MagicBrush benchmark in Table 4. Here are our main observations: 1) Compared to canonical image editing baselines like HIVE, SD-SDEdit, and IP2P (zero-shot), merely training an editing diffusion model on the free-form editing data of ULTRAEDIT (**downsampled** to ~450K to match the training set size of IP2P, denoted as *Ours, trained w/o region data*) already attains significant improvement over the baseline, confirming the advantages brought by our dataset to general image editing; 2) When also considering the relatively small-scale region-based editing data (~100K region-based + ~350K free-form, denoted as *Ours*) and evaluate on the same setting without editing region input, the general editing performance can be boosted considerably, verifying the effectiveness of region-based editing data to image editing in general; 3) Finally, when being evaluated using region input, the model trained on both free-form and region-based editing data (still downsampled to ~450K in total) sets the new record on MagicBrush especially on the challenging multi-turn setting, demonstrate that our region-based editing data can indeed help with the emergence of region-based editing capability, while existing approaches that also utilize masks (GLIDE, Blended Diffusion) are poor at effectively guiding their editing with region input.

Table 5: **Results on Emu Edit Test.** We present the benchmark results for various methods and models trained on different scales of data.

| Method | CLIPdir↑ | CLIPout↑ | L1↓ | CLIPimg↑ | DINO↑ |
|---|---|---|---|---|---|
| InstructPix2Pix (450K) | 0.0784 | 0.2742 | 0.1213 | 0.8518 | 0.7656 |
| MagicBrush (450+20K) | 0.0658 | 0.2763 | 0.0652 | 0.9179 | **0.8924** |
| Emu Edit(10M) | 0.1066 | **0.2843** | 0.0895 | 0.8622 | 0.8358 |
| Ours (450k, w/o region data) | 0.0823 | 0.2778 | 0.0626 | 0.8617 | 0.8190 |
| Ours (1M w/o region data) | 0.0862 | 0.2804 | **0.0515** | **0.8915** | 0.8656 |
| Ours (1.5M, w/o region data) | 0.0952 | 0.2808 | 0.0600 | 0.8659 | 0.8243 |
| Ours (2M, w/o region data) | 0.0960 | 0.2811 | 0.0608 | 0.8689 | 0.8269 |
| Ours (2.5M, w/o region data) | 0.0997 | 0.2822 | 0.0854 | 0.8407 | 0.7814 |
| Ours (3M, w/o region data) | **0.1076** | 0.2832 | 0.0713 | 0.8446 | 0.7937 |

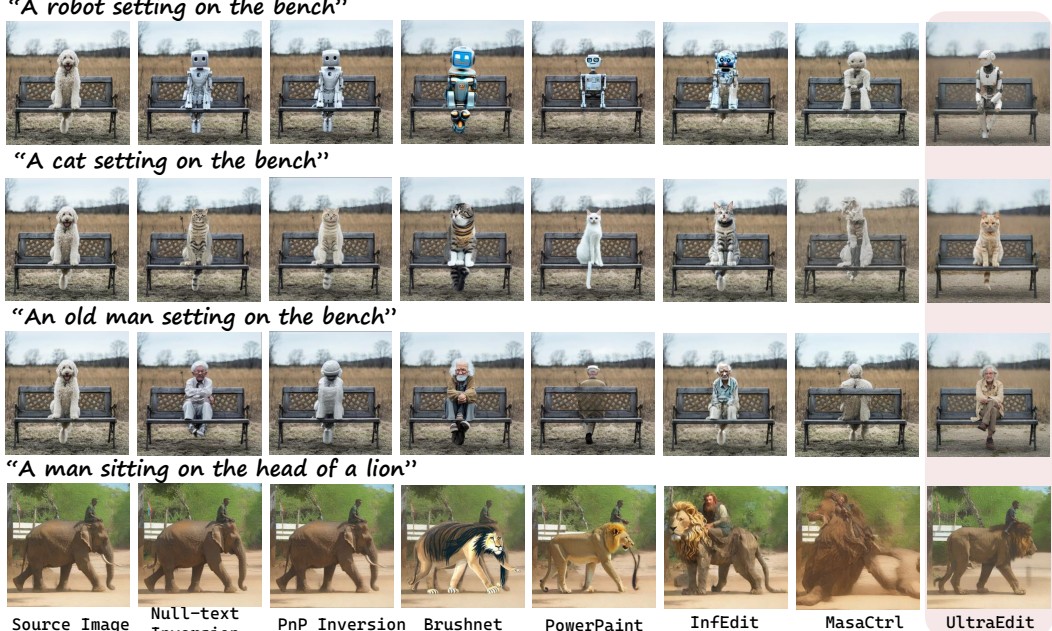

Figure 4: Qualitative comparison of images generated by our data generation pipeline with other zero-shot image editing methods.

**Note:** As also reported in [60], we found the MagicBrush benchmark itself introduces biases towards its training set, leading to unfairly high results of models trained on its training set, *i.e.* IP2P w/ MagicBrush. It effectively overfits on MagicBrush and loses general editing ability on other datasets. Poor results of the same model on Emu Edit (*MagicBrush* in Table 5) verify this.

### 3.3 Main Result II: Scaling Effect of ULTRAEDIT on Emu Edit Test

In Table 5, we present results on how progressively scaling the size of ULTRAEDIT could affect the performance of general image editing on the challenging Emu Edit Test. We made the following observations: 1) In general, models trained on ULTRAEDIT attain better results than IP2P, and the advantages expand significantly when the scale of the dataset increases; 2) Scaling effect can be more significant on metrics indicating editing, *i.e.*, CLIPdir (measuring the if the editing is consistent with the changes between the captions of source and target images) and CLIPout (measure if the edited image is consistent with the caption of the target image), and we ultimately set a new record on Emu Edit than baseline trained on a proprietary 10M dataset; 3) When it comes to content preserving metrics, *i.e.*, L1, CLIPimg, and DINO, our hypothesis to the trend is: as the data scale increases, the model is gradually learning to make hard edits, therefore it only starts to edit more after a certain scale of data is reached; 4) Compared to Emu Edit baseline, our model can produce both accurate

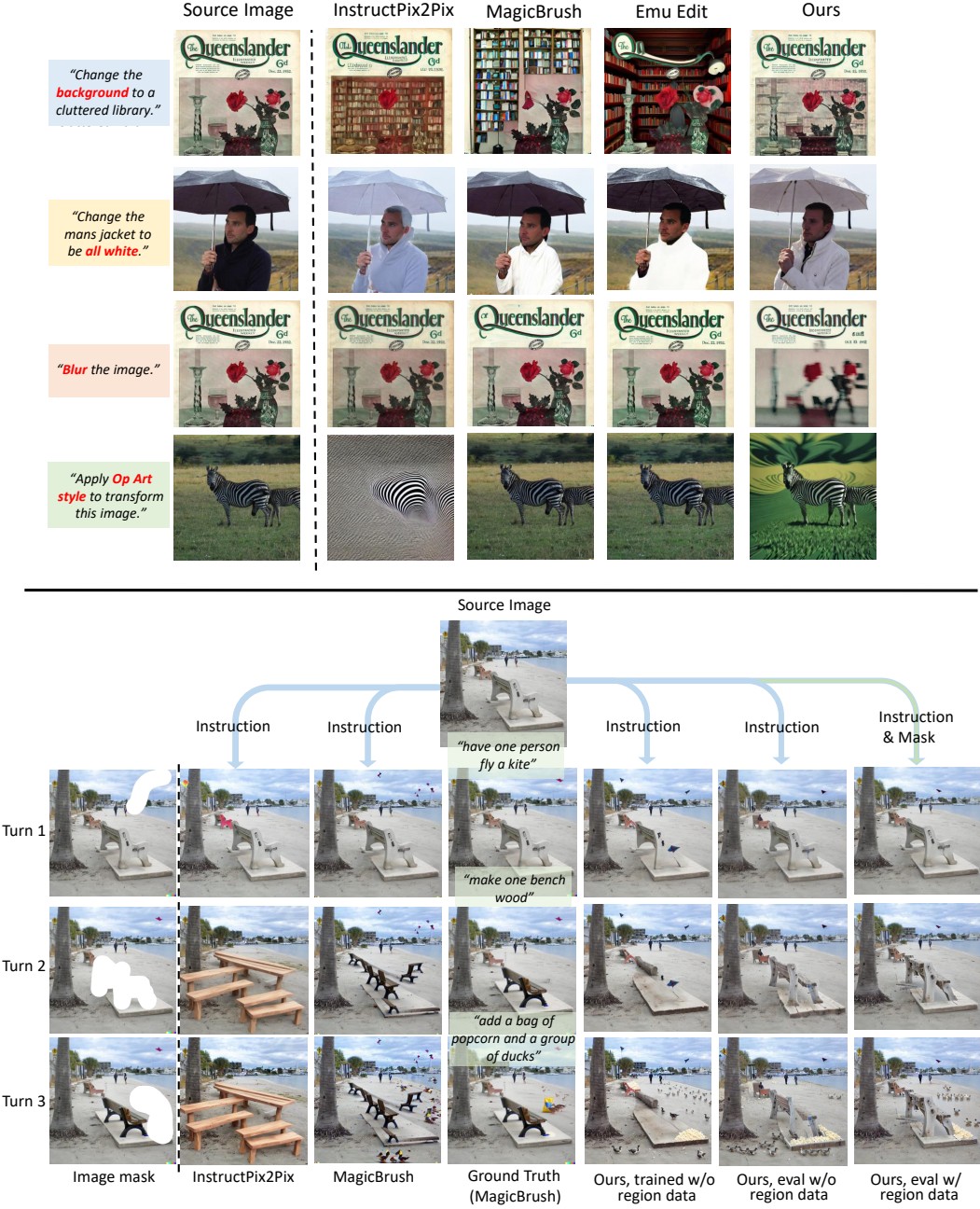

Figure 5: **Qualitative evaluation.** (Top) On four distinct tasks on Emu Edit, listed from top to bottom: background, color, global, style; (Bottom) Multi-turn editing on MagicBrush.

edits (hight CLIPdir and CLIPout) while preserving the content of the original image (lower L1), indicating that our model is mostly editing the image while considering the context, not just creating new contents. Our qualitative results (Figure 5) provide further evidence on this.

## 3.4 Qualitative Evaluation

**Data Generation.**    In Figure 4, we present qualitative examples generated by our data generation pipeline alongside various image editing methods for comparison. We evaluate our pipeline against several baselines, including image inversion methods like Null-text Inversion [42] and PnP Inversion [27]; inpainting methods such as BrushNet [28] and PowerPaint [76]; and zero-shot image editing methods like InfEdit [69] and MasaCtrl [12]. Our data generation pipeline, leveraging SDXL-turbo, requires only 1-4 steps to produce a sample, which is significantly faster than other methods by

Table 6: **Ablation study on real image anchors.** The first three rows present results for models trained on different scales from the ULTRAEDIT. The last three rows show results for models trained on data from the same generation pipeline but without using the real image as an anchor.

| Data Type | Data Volume | CLIPdir↑ | CLIPimg↑ | CLIPout↑ | L1↓ | DINO↑ |
|---|---|---|---|---|---|---|
| UltraEditing | 450k | 0.0823 | 0.8617 | 0.2778 | 0.0626 | 0.8190 |
| | 1M | 0.0925 | 0.8696 | 0.2807 | 0.0599 | 0.8307 |
| | 1.5M | 0.0952 | 0.8659 | 0.2808 | 0.0600 | 0.8243 |
| w/o image anchor | 450k | 0.0728 | 0.8716 | 0.2796 | 0.0848 | 0.8154 |
| | 1M | 0.0638 | 0.8837 | 0.2770 | 0.0674 | 0.8353 |
| | 1.5M | 0.0720 | 0.8643 | 0.2781 | 0.0714 | 0.8105 |

magnitudes. Importantly, this efficiency does not compromise the quality. Our pipeline demonstrates competitive performance against other baselines, highlighting the quality of ULTRAEDIT.

**Image Editing.**    In Figure 5, we provide some qualitative examples of different editing tasks on Emu Edit Test and multi-turn editing on MagicBrush generated by the Stable Diffusion v1.5 trained on ULTRAEDIT. More examples can be found in Appendix D. Our main observations are: 1) most baselines, especially Emu Edit, tend to *overedit* the image, *i.e.* creating content while ignoring the context of the source image. For example, Emu Edit makes crude modifications by setting the entire person's body as blank. We attribute these shortcomings to the biases introduced during the construction of Emu Edit's training data; 2) Many baselines also fail to generalize to novel instructions, *e.g.*, blurring, adding special style, *etc.*; 3) For multi-turn editing, even the MagicBrush baselines cannot complete the long-term editing coherently, while our model, even without editing region input, can strictly follow the instruction, *e.g. one person*, *one bench*, etc, and reaches the best results with region information, which confirms the effectiveness of region-based data in ULTRAEDIT.

### 3.5    Insights and Analysis

In this section, we investigate how two of our designs when curating ULTRAEDIT: *real image anchors* and *region-based editing* could affect the performances. All results are conducted on Emu Edit Test.

**Real Image Anchors.**    We conduct an ablation study to verify the effectiveness of incorporating real images as anchors during data generation. We train the editing model using two distinct datasets: free-form image editing data from ULTRAEDIT and the editing data generated using the same pipeline of ULTRAEDIT without using real images as anchors, *i.e.* identical to the data creation pipeline of InstructPix2Pix [10]. The models were trained on data volumes of 450K, 1M, and 1.5M for both datasets. The results can be found in Table 6. Our key observations: 1) Dataset generated with real image anchors generally leads to better models across all three scales; 2) The scaling effect only presents when real image anchors are adopted. We hypothesize that datasets without them could suffer from more severe image biases and therefore hinder the effect of further scaling up. In Appendix D.3, we demonstrate more qualitative results comparing the image editing generation method with and without using real images as anchors.

**Free-from vs. Region-based Editing.**    We then explore the impact of incorporating region-based editing data during model training. We experiment with varying amounts of free-form editing data, ranging from 200K to 400K instances, and region-based editing data, ranging from 30K to 90K instances. We demonstrate the performances on Emu Edit Test of models trained on the possible volume combinations of these two types of data in Figure 6. It can be seen that: 1) Region-based editing data, despite its relatively smaller scale, can help with free-form editing tasks; 2) While scaling free-form editing data has a significant impact on the performance of region-based editing, we need to ensure a considerable volume of region-based data to ensure peak results. In Appendix D.4, we demonstrate more qualitative results with region input and the model exhibits significantly more precise

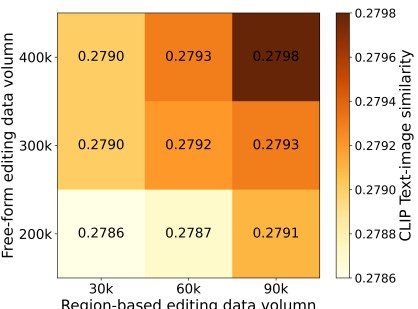

Figure 6: **Ablations on region-based editing**. We report CLIPout as the main metric.

operations for background and localized edits, validating the effectiveness of the incorporation of the region-based editing.

# 4    Related Work

**Image Editing via Generation.**    Editing real photos according to specific instructions has long been a notable task in image processing [21, 16, 39, 72, 54, 45]. Powerful Large-scale diffusion models have significantly facilitated text-based image editing [29, 55, 33, 14, 40]. SDEdit [41] applies noise to the guidance image at an intermediate diffusion step and then denoises using the target description. Prompt-to-Prompt [24] injecting the input caption attention maps to the target ones. Null-Text Inversion [42] inverts the source image to the null-text embedding for editing, eliminating the need for original captions. Plug-and-Play [64] incorporates spatial features besides attention maps for better global image editing. Imagic [29] supports complex textual instructions via text embedding optimization and model fine-tuning. GLIDE [44] and Imagen Editor [67] fine-tuning the model to take channel-wise concatenation of the input image and mask. Blended Diffusion [5] blends the input image in the unmasked regions in the diffusion step. Meanwhile, instruction-based image editing has been introduced as a user-friendly method for image editing. Instructpix2pix [10] and HIVE [73] both are trained on generated editing data to handle user-written instructions during inference. MagicBrush [71] creates a manually-annotated dataset for fine-tuning Instructpix2pix. Emu Edit [60], trained on 10 million *proprietary* multi-task data, demonstrates state-of-the-art performance.

**Image Editing Dataset.**    Table 1 compares various image editing datasets, revealing that high-quality data are scarce and challenging to obtain. The largest dataset only contains around 300,000 samples. EditBench [68] is manually curated with only 240 examples. MagicBrush [71]manually annotated by hearing online labors using DALL-E 2 [50] for crafting dataset, also limited in dataset size. For automatically annotated datasets, InstructPix2Pix utilizes prompt-to-prompt [24] to generate data pairs based on the captions form LAION-Aesthetics [59] and edited captions generated by GPT-3 [11], but biases in the generative model and insufficient information in web-crawled captions make the generated image samples fail to represent the editing instructions, as shown in the Table 1. The the limited scope of the image instructions types of InstructPix2Pix further limits the utility of the dataset. HQ-Edit[26] uses advanced models like GPT-4 [2, 1] and DALL-E 3 [6] for generating image editing pairs, but may fail to preserve fine-grained details and realism in the target image.

# 5    Conclusion

We've presented ULTRAEDIT, a large-scale, high-quality dataset for instruction-based image editing. We mitigate the issues in existing editing datasets with a *systematic* approach for automatic data generation: combining LLM creativity and in-context examples from human raters for more diverse editing instructions; the use of real images as anchors for more balanced generations; support of region-based editing via automatic region extraction. Experiments on challenging MagicBrush and EmuEdit benchmarks confirm the effectiveness of training on our dataset. Possible future work includes further expanding the region-based editing data and bootstrapped training for image editing.

## Acknowledgement

We extend our heartfelt gratitude to the anonymous reviewers whose dedication and insightful feedback have significantly enhanced the caliber of this paper. Their constructive critiques and valuable suggestions were instrumental in refining our work. Additionally, we are deeply appreciative of the Program Chairs and Area Chairs for their meticulous handling of our submission and for their comprehensive and invaluable feedback. Their guidance has been pivotal in elevating the quality of our research.

This work is supported by the National Science Foundation of China under Grant No.61936012 and 61876004.

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

Table 7: Datasets used to form the high-quality image caption dataset.

| Dataset | #Samples | License | Annotator |
|---------|----------|---------|-----------|
| MS COCO [36] | 164,000 | CC BY 4.0 | Human |
| Flickr [70] | 31,783 | Custom | Human |
| NoCaps [4] | 45,000 | CC BY 2.0 | Human |
| VizWiz Caption [8] | 23,431 | CC BY 4.0 | Human |
| TextCaps [62] | 28,408 | CC BY 4.0 | Human |
| Localized Narratives [47] | 849,000 | CC BY 4.0 | Human |
| ShareGPT4V [13] | 1,200,000 | CC BY-NC 4.0 | GPT-4V |
| LAION-LVIS [58] | 220,000 | Apache-2.0 | GPT-4V |

# A  Implementation Details

## A.1  Collection of High-Quality Image Caption Data

To ensure the diversity and quality of the image editing pairs in ULTRAEDIT, our data generation pipeline relies on high-quality oracle data to mitigate biases within the generation models. Consequently, we focus on building the image editing dataset based on real images and their captions, enhancing the dataset's reliability in real-world scenarios and providing more comprehensive guidance for data generation than text-only data.

Like previous works [34, 13, 71, 75], we gathered source data from various public data sources with image caption data, as illustrated in Table 7, which includes diverse images with either manually annotated captions or detailed captions generated by advanced image captioning models [13, 58]. After filtering out images with excessively long or short captions, our collection amounted to 1.6 million high-quality image-caption pairs, which will be used for edit instruction generation and subsequent image generation.

## A.2  Instruction and Caption Generation

To obtain high-quality editing instructions, we introduce a pipeline that combines human raters and language models for generating edit instructions and corresponding captions for subsequent image generation. Large language models have demonstrated remarkable abilities in various areas, such as agents [18, 25, 61, 35] and tool uage [57]. In our practice, we utilize the LLM to generate suitable image editing instructions. Firstly, we use language models to expand manually crafted edit examples to a set of 100,000 examples, as shown in Table 8. These examples serve as in-context learning examples to help the language models grasp the understanding of editing styles and requirements, enabling them to generate suitable editing instructions and corresponding edited caption. The query prompt is illustrated in Table 9.

We sample 50 editing instructions and 10 edit examples as in-context learning examples for querying the language model. The language model then generates appropriate editing instructions and corresponding edited captions for the given image captions from the collection. Leveraging the in-context-learning capabilities and generalization ability of the language model, we ultimately generate 4.16 million text-only data comprising creative yet sensible edit instructions and corresponding captions. Each case consists of a high-quality image caption for a real image, an editing instruction, and an edited caption corresponding to a target image.

## A.3  Region-based Data Generation

For generating region-based image editing data, we first employ the recognize-anything [74] to identify objects in the source images. We then query the language model with the obtained object lists, edit instructions, and corresponding image captions to determine the target objects of the editing instruction using the query prompt in Table 10. If the editing instruction is object-oriented, the language model identifies the objects involved in the editing; otherwise, the entire image is considered as the editing target.

To generate region-based edited images, we use the Grounding DINO [38] to obtain bounding boxes of editing areas in real images, serving as coarse-grained masks. Subsequently, we perform SAM [30] on these bounding boxes to derive fine-grained object masks, expanding them to create contour masks

Table 8: Examples of instructions and their corresponding captions after Expansion.

| Original Caption | Edit Instruction | Edited Caption |
|---|---|---|
| Two Asian dolls with big noses, fancy purple dresses, and golden hats. | Replace the dolls with miniature elephants in colorful traditional Indian cloth | Two miniature elephants wearing colorful traditional Indian cloth. |
| A man is watching TV in bed with only his foot showing. | Remove the bed and place the man in a deserted beach scene. | A man is watching TV on a deserted island beach with only his foot showing. |
| a person throwing a red frisbee, which is currently in mid-air, slightly to the right and above the person's hand. | Change the color of the frisbee to blue | a person throwing a blue frisbee, which is currently in mid-air, slightly to the right and above the person's hand. |
| A slipper near the edge of a concrete floor near small rocks. | Transform the slipper into a glass one | A glass slipper near the edge of a concrete floor near small rocks. |
| A woman wearing a shirt for the Religious Coalition for Reproductive Choice. | Change the background to a bustling cityscape at night | A woman wearing a shirt for the Religious Coalition for Reproductive Choice in front of a bustling cityscape at night. |
| A pot and some trays are in a kitchen. | Add a warm, inviting atmosphere to the image | The warm glow highlights a pot and some trays in a cozy kitchen. |
| a person that is jumping his skateboard doing a trick. | Turn the skateboard into a flying carpet | a person that is jumping his flying carpet doing a trick. |
| A stuffed bear is hanging on a fence. | Make it a snowy winter landscape | A stuffed bear is hanging on a snowy winter landscape fence. |
| A police dog wearing his bullet proof vest. | replace the background with a city skyline | A police dog wearing his bullet proof vest in a city skyline. |
| A baseball game is going with children playing a runner is about to hit the base. | Turn the baseball field into a magical forest | Children playing a runner is about to hit the base in a magical forest. |
| A small horse carries a women in a sled. | Turn the horse and sled into a spaceship traveling through outer space | A futuristic spaceship travels through outer space. |
| a person wearing a life jacket participating in water sports like water skiing. | Add a family of dolphins swimming around the person in the water | a person wearing a life jacket participating in water sports like water skiing, with a family of dolphins swimming around him in the water. |

that define the editing region. During image generation, we have observed irregular color boundaries between the editing region and the rest of the image. To ensure smooth transitions and high image quality, we fuse the fine-grained and bounding box masks to create a soft mask guiding the image generation. Specifically, for the editing region latent $M_f$ and bounding box latent $M_b$, we fuse the two masks, making the region between them a soft mask region $M_s$. The generation pipeline can be formulated as follows:

$$z_{t-1} = \begin{cases} (1 - M_s) \cdot z_T + M_s \cdot DM(z_t) & \text{if } t \mod 2 == 0 \\ DM(z_t) & \text{otherwise} \end{cases} \quad (2)$$

Additionally, we define $M_s$ as:

Table 9: Query prompts for LLMs to write edit instructions and corresponding captions.

**Prompt for Writing Edit Instruction**

| Element | Content |
|---|---|
| Intro | I will present a series of image editing instruction examples essential for mastering and understanding a variety of editing styles and requirements. Here are some sample instructions: |
| Instruction Examples | {instruction_str} |
| Task Description | I will provide one image caption corresponding to a specific image. You are required to apply the learned editing techniques to form suitable, detailed, and accurate editing instructions for the image defined by the caption. Note that your editing instructions should be distinct from the examples provided. Then, produce a description corresponding to the revised image after applying the editing instruction. Only necessary amendments should be made for the new image caption. |
| Output Format | The output format should be "original image caption; edit instruction; new image caption". Maintain the given format for the result. Please ensure to deliver solely the result, without incorporating any additional titles. |
| Image Caption | The image caption is: {caption} |
| Produce Instances | Produce three suitable instances based on the caption and return the list. |
| Examples | Here are some output examples for your reference: {example_str} |
| Response | Response: |

$$M_s = \begin{cases} s & \text{for elements in } M_b \setminus M_f \\ M_f & \text{otherwise} \end{cases} \tag{3}$$

where $M_f$ is the editing region latent, $M_b$ is the bounding box latent, and $s$ is the hyperparameter that determines the inpainting rate. During the generation, During the generation, we set $s$ to range from 0.2 to 0.8.

## A.4 An Improved Baseline for Free-form and Region-based Image Editing

We fine-tune the Stable Diffusion 1.5 model [52] using the Diffusers library [65] with data from ULTRAEDIT. We maintain the hyperparameters as set in Brooks et al. [10]. Specifically, we train the model on $8 \times 80\text{GB}$ NVIDIA A100 GPUs with a total batch size of 256. Following prior works [10, 71], we use an image resolution of $256 \times 256$ for training and $512 \times 512$ for generation.

To incorporate additional guidance from region masks, we concatenate the latent of the Region Mask $M_s$ with the noisy latent $Z_T$ and the latent of the source image $Z_I$ to form the input to the diffusion model. We add four additional channels to the UNet of the diffusion model to accommodate the latent of the region mask $M_s$. The weights of the UNet are initialized with the pretrained diffusion model, while the extra eight channels (four for the source image latent $Z_I$ and four for the mask latent $M_s$) in the convolutional layers of the diffusion UNet are randomly initialized. The model is then trained using a mixture of free-form and region-based image editing data from ULTRAEDIT. For free-form image editing data, the model takes a blank mask as input to implicitly indicate that the editing should affect the entire image.

Table 10: Query prompts for LLMs to capture objects that need editing.

**Prompt for Capturing Editing Object**

| Element | Content |
|---|---|
| Intro | The following prompt provides an instruction for image editing, an original image caption, a revised image caption that reflects the given edit instruction, and a set of objects detected by an object detection algorithm. |
| Edit Instruction | Edit Instruction: "{edit_instruction}" |
| Original Caption | Original Image Caption: "{input_text}" |
| Revised Caption | Revised Image Caption: "{output_text}" |
| Object List | Set of Objects Identified by the Recognition Model: {object_list} |
| Task Description | Your task is to identify the objects most likely to be modified based on the information provided above. Consider this from a comprehensive perspective; note that some objects might not be explicitly mentioned in the instructions or the Identified Objects list, but their appearance could still be affected. Please use precise words or phrases in your response. |
| Note | Note:
1. If you can't identify any specific edited object (e.g., a style transfer involving the entire image instead of a single object; add/move an object, which does not fit object-oriented editing instructions), please respond with "NONE".
2. Your response should exclusively identify the objects requiring edits, excluding any extra context or details.
3. Please list the objects to be edited, separating each one with a comma. The number of objects identified in the answer should not exceed 2. |
| Response | Response: |

When training the model exclusively with Free-form Image Editing data, we strictly follow the settings of Brooks et al. [10] without making any additional modifications.

# B Statistics of ULTRAEDIT

Table 11: Statistics of Free-form and Region-based Image Editing Data. The table shows the instance numbers, number of unique instructions, and their respective proportions for different instruction types.

| Data Type | Statistic | Change | | | Transform | | Add | Replace | Turn | Others | Total |
|---|---|---|---|---|---|---|---|---|---|---|---|
| | | Color | Global | Local | Global | Local | | | | | |
| Free-form | Inst. No. | 111,563 | 204,294 | 500,108 | 150,851 | 597,165 | 909,065 | 683,529 | 490,219 | 353,289 | 4,000,083 |
| | Proportion (%) | 2.79 | 5.11 | 12.50 | 3.77 | 14.93 | 22.73 | 17.09 | 12.26 | 8.83 | / |
| | Unique Inst. | 27,436 | 24,020 | 92,891 | 26,587 | 117,063 | 114,647 | 133,222 | 102,280 | 86,180 | 724,326 |
| | Proportion (%) | 3.79 | 3.32 | 12.82 | 3.67 | 16.16 | 15.83 | 18.39 | 14.12 | 11.90 | / |
| Region-based | Inst. No. | 2,912 | 3,515 | 15,774 | 2,796 | 21,807 | 11,918 | 25,749 | 16,628 | 7,080 | 108,179 |
| | Proportion (%) | 2.69 | 3.25 | 14.58 | 2.58 | 20.16 | 11.02 | 23.80 | 15.37 | 6.54 | / |
| | Unique Inst. | 1,056 | 727 | 5,032 | 762 | 6,835 | 3,201 | 8,064 | 5,256 | 2,620 | 33,553 |
| | Proportion (%) | 3.15 | 2.17 | 15.00 | 2.27 | 20.37 | 9.54 | 24.03 | 15.66 | 7.81 | / |

In this section, we dive into the characteristics and statistics of ULTRAEDIT. We present ULTRAEDIT, a large-scale, diverse, and high-quality real-image-based image editing dataset designed to advance the capabilities of image editing models. ULTRAEDIT comprises over 4,000,000 instruction-based free-form image editing instances and 100,000 region-based image editing instances, making it the largest open-source image editing dataset. Notably, it is also the first large-scale dataset focused on region-based image editing. Table 11 illustrates the statistics of the image editing data in ULTRAEDIT.

It shows the numbers and proportions of the different instruction types and data types of ULTRAEDIT. Moreover, the Figure 7 shows the distribution of keywords in the instructions of ULTRAEDIT for various instruction types.

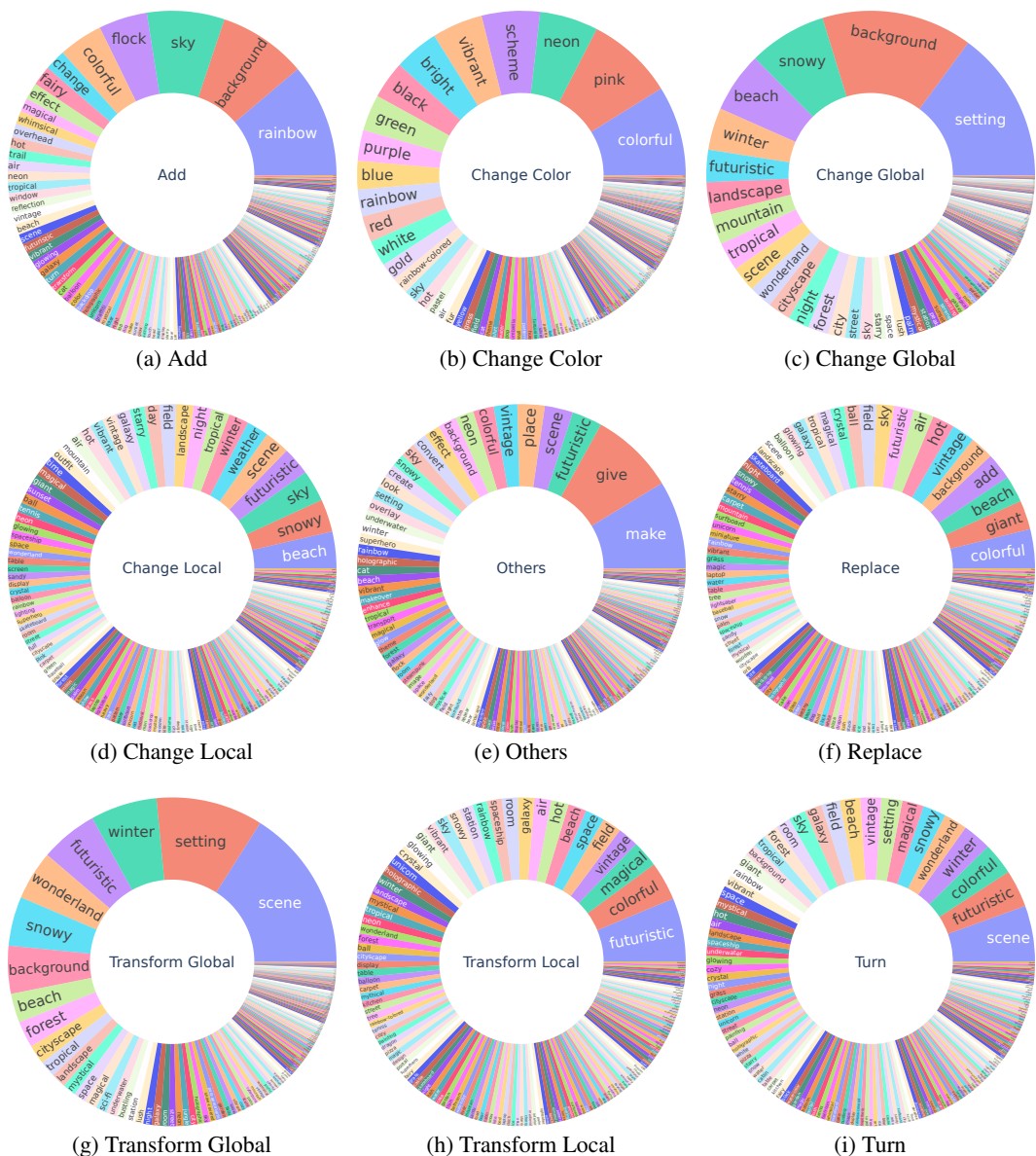

Figure 7: Distribution of keywords in the instructions of ULTRAEDIT for various instruction types

## B.1 Comparison with other dataset

In this section, we compare our dataset with the InstructPix2Pix (IP2P) dataset in Free-form image editing. We report the results of automatic metrics to evaluate the data quality of each dataset.

As illustrated in Table 12, our dataset outperforms the InstructPix2Pix (IP2P) dataset across all tasks. Notably, the higher CLIPimg scores observed in some tasks for the IP2P dataset suggest that while the image pairs in the dataset may exhibit semantic similarity, it does not achieve the desired visual similarity, which is crucial for successful image editing. This highlights a fundamental shortcoming in the IP2P dataset and underscores the superior of our dataset.

Table 12: Comparison of dataset quality between ULTRAEDIT and InstructPix2Pix (IP2P) using automatic metrics. We evaluates the data quality across different tasks types.

| Task | Dataset | CLIPin | CLIPout | CLIPdir | CLIPimg | SSIM | DINOV2 |
|---|---|---|---|---|---|---|---|
| Change | Ours | 0.2849 | 0.3024 | 0.2967 | 0.8441 | 0.6360 | 0.7403 |
| | IP2P | 0.2667 | 0.2661 | 0.2317 | 0.8557 | 0.5685 | 0.6194 |
| Transform | Ours | 0.2851 | 0.3005 | 0.2902 | 0.8289 | 0.6251 | 0.6875 |
| | IP2P | 0.2646 | 0.2680 | 0.1974 | 0.8667 | 0.5853 | 0.6972 |
| Turn | Ours | 0.2846 | 0.3018 | 0.2922 | 0.8321 | 0.6255 | 0.6949 |
| | IP2P | 0.2654 | 0.2698 | 0.2015 | 0.8575 | 0.5526 | 0.6419 |
| Add | Ours | 0.2786 | 0.3145 | 0.2957 | 0.8661 | 0.6645 | 0.7758 |
| | IP2P | 0.2629 | 0.2744 | 0.1990 | 0.8851 | 0.6318 | 0.7026 |
| Others | Ours | 0.2843 | 0.3038 | 0.2981 | 0.8374 | 0.6420 | 0.7048 |
| | IP2P | 0.2657 | 0.2706 | 0.1929 | 0.8629 | 0.5734 | 0.6847 |
| Overall | Ours | 0.2834 | 0.3049 | 0.2950 | 0.8427 | 0.6401 | 0.7231 |
| | IP2P | 0.2650 | 0.2694 | 0.1982 | 0.8660 | 0.5826 | 0.6859 |

## B.2 More Examples of ULTRAEDIT

In this section, we showcase additional examples from ULTRAEDIT to illustrate the versatility and robustness of our dataset in various image editing tasks. The free-form editing data is depicted in the left two columns, while the region-based image editing data examples are in the right column. The examples highlight both Free-form and Region-based editing capabilities. It can be noticed that, due to using real images as anchors, our data shows high diversity in real-world scenarios, including text, natural environments, human figures, abstract objects, and even blurred low-quality images.

In Figure 8 and Figure 9, editing examples not only contain text modification, and abstract object editing, but also multi-step editing within a single instruction and fine-grained editing. Moreover, because of high-quality captions derived from open-source image caption datasets for generating editing instructions, the generated instructions are highly related to the source image. The region-based image editing data demonstrates high image element preservation in the editing examples. For instance, in the examples in the right column, the target images only perform edits within the masked area and keep the rest unchanged, even for highly blurred texts and human facial expressions in the figure.

## C  Baseline and Metrics

### C.1  Baselines.

We set the following models as baselines, categorized into instruction-based image editing methods and global description-guided image editing methods, the latter requiring global descriptions of the target image to perform zero-shot editing. The instruction-based image editing methods include: InstructPix2Pix [10], HIVE [73], MagicBrush [71], and Emu Edit [60]. The global description-guided image editing methods include: Null Text Inversion [42], SD-SDEdit [41], GLIDE [44], and Blended Diffusion [5]. Notably, GLIDE and Blended Diffusion require a mask for editing.

**Instruction-Based Methods:**

- **InstructPix2Pix** uses automatically generated instruction-based image editing data to fine-tuning Stable Diffusion [53] and performance image editing based on the instructions during the inference, without any test-time tuning.

- **HIVE** is trained with more data similarly to InstructPix2Pix, and is further fine-tuned with a reward model trained on human-ranked data.

- **MagicBrush:** is a variant of InstructPix2Pix, which is fine-tuned on the human-annotated dataset, MagicBrush.

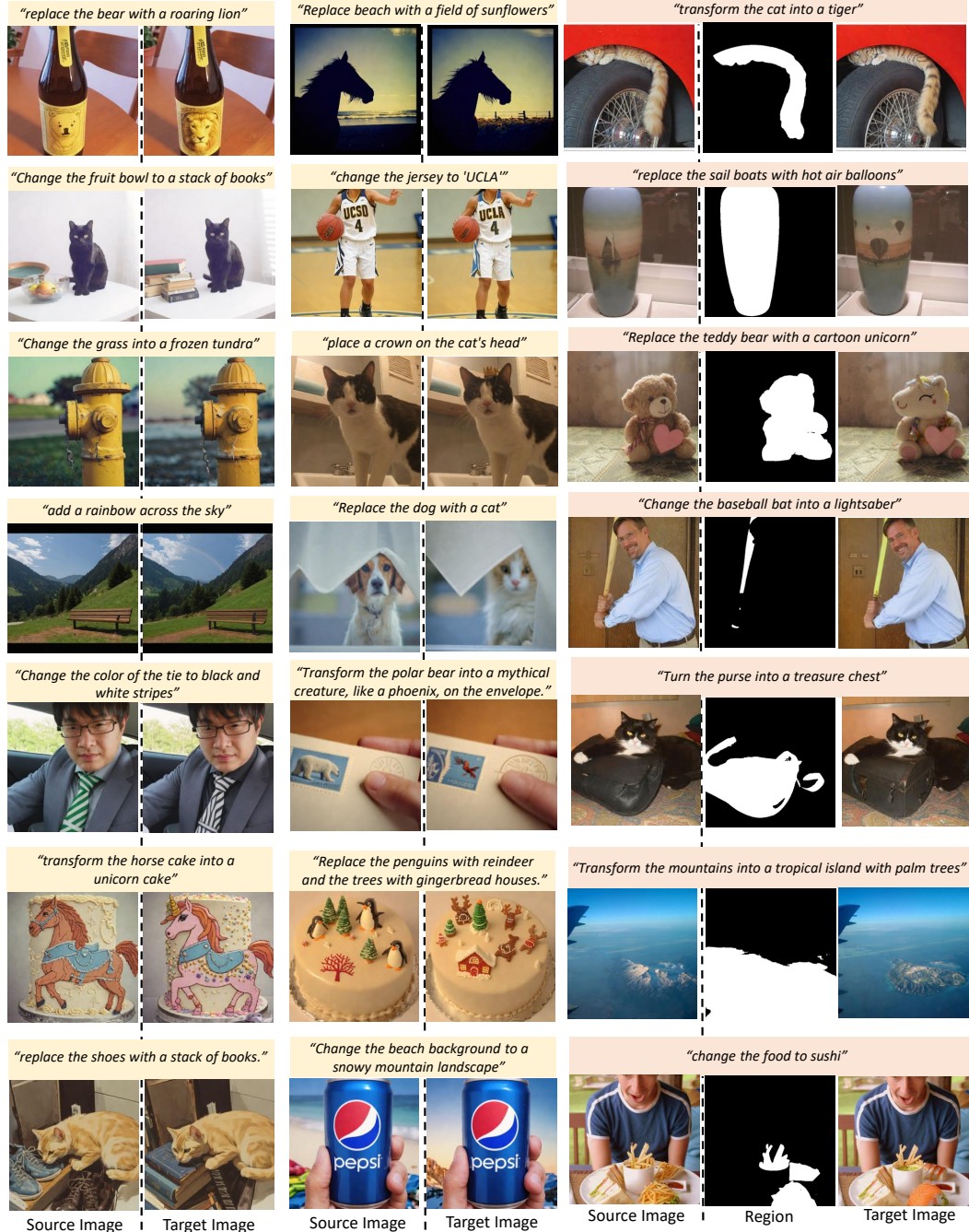

Figure 8: **More Examples of ULTRAEDIT**. Free-form (left) and region-based (right) image editing.

- **Emu Edit:** is a closed-source model that supports multi-task image editing and achieves state-of-the-art performance. It is trained on a diverse set of tasks using 10 million of training data, including image editing and computer vision tasks.

**Global Description-Guided Methods:**

- **Null Text Inversion:** inverts the source image with DDIM [63] trajectory and then performs editing during the denoising process with text-image cross-attention control [24].

- **SD-SDEdit:** noises the guidance image to an intermediate diffusion step, and then denoises it using the target description.

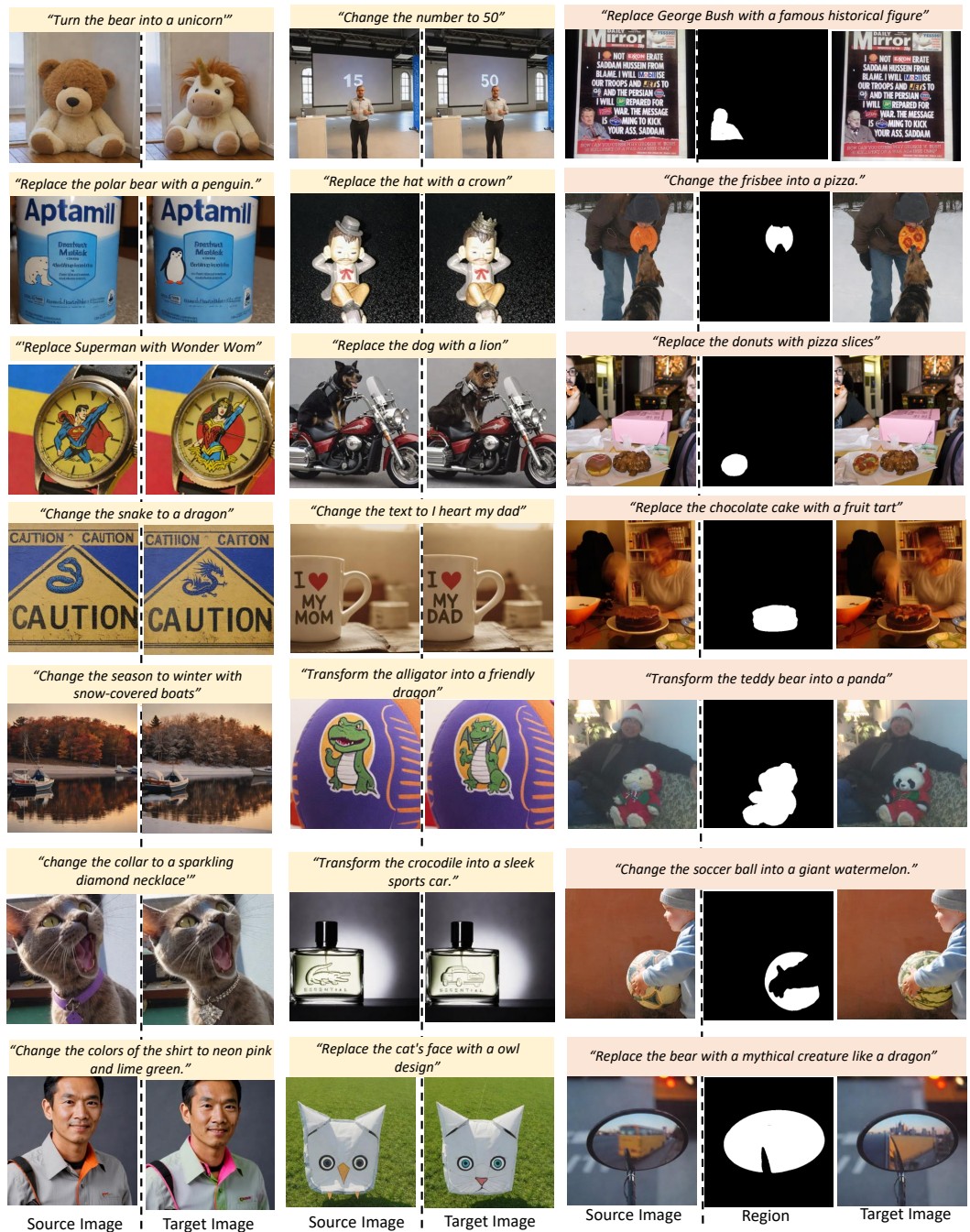

Figure 9: **More Examples of ULTRAEDIT**. Free-form (left) and region-based (right) image editing.

- **GLIDE:** is trained with 67M text-image pairs to fill in the masked region of an image conditioned on the local description with CLIP guidance.
- **Blended Diffusion:** blends the input image in the unmasked regions with the context in the noisy source image during each denoising timestep to enhance region-context consistency.

## C.2 Details on Benchmarks and Metrics

MagicBrush aims for evaluating the single and multi-turn image editing ability of the model. It provides annotator defined instructions and editing masks, as well as the ground truth images generated by DALLE-2 [50] for evaluation, allowing for more effective metric assessment of the

|  | Ours w/ UltraEdit | MagicBrush [71] | InstructPix2Pix [10] |
|---|---|---|---|
| TrueSkill score | $25.5 \pm 0.8$ | $23.7 \pm 0.7$ | $22.6 \pm 0.7$ |

Table 13: TrueSkill [23] scores of image editing models evaluated by human raters on MagicBrush [71] test set.

model's editing performance. However, this dataset also suffers from inherent bias. During data collection, annotators were directed to use the DALLE-2 image editing platform to generate the edited images. Thus, this benchmark is biased towards generated images and editing instructions that the DALLE-2 editor can successfully follow, which may compromise both its diversity and complexity. Following the setting of MagicBrush [71], we utilize L1 and L2 to measure the pixel-level difference between the generated image and ground truth image. And also adopts the CLIP similarity and DINO similarity to measure the overall similarity with the ground truth. Finally, the CLIP-T is used to measure the text-image alignment between local descriptions and generated images CLIP embedding.

Emu Edit Test aims for reducing bias of the annotator defined dataset and reach higher diversity. It contains the devise relevant, creative, and challenging instructions and high quality captions that capture both important elements in the image for source and target images, without any ground truth images. Consequently, consistent with the Emu Edit [60], we utilize the L1 distance, CLIP image similarity and DINO similarity between the **source images** and **edited images** to measure the the model's ability of preserving elements from the source image. Also, we use the CLIP text-image similarity between edited image and output caption and the CLIP text-image direction similarity(CLIPdir) to measure the instruction following ability of the model. Specifically, the CLIPdir measures agreement between change in caption embedding and the change in image embedding.Since the Emu Edit [60] does not specify the versions of the CLIP and DINO models used for the metric, we adopted the settings utilized by MagicBrush to maintain alignment with other benchmarks. Specifically, the versions are `ViT-B/32` for CLIP and `dino_vits16` for DINO embeddings. We ensure consistency by rerunning all results of different methods on Emu Edit benchmark. Additionally, there are known issues with the quality of the benchmark, wherein some image-caption pairs appear incorrect. These issues include placeholder captions (e.g., 'a train station in city') or instances where source and target captions are identical. To address these problems, we simply remove the incorrect cases prior to evaluation. Despite the Emu Edit Test eliminating bias and overfitting at the image level by not providing ground-truth images, the evaluation metrics still implicitly measure the model's editing ability.

## D   Qualitative and Human Evaluations

### D.1   Human Evaluation

We conducted human evaluations to assess the consistency, instruction alignment, and image quality of the edited images generated by our model trained on UltraEdit using the MagicBrush benchmark and Emu test benchmark. We first compared the performance of our model with the MagicBrush [71] and instructPix2Pix [10] models through a comprehensive human evaluation on MagicBrush benchmark. Additionally, we compared the performance of various models trained using our dataset with Emu Edit [60] on the Emu test benchmark. For the two evaluations, we randomly sampled 500 examples from the test sets of the MagicBrush benchmark and the Emu test benchmark, respectively.

For each sample, the evaluators compared the consistency, instruction alignment, and image quality of the edited images generated by the different models. As shwon in Figure 10, the evaluators were asked to determine which edited image was better by selecting between "First Image", "Second Image", or "Tie". The results are evaluated with TrueSkill [23] rating system. The scores of these evaluations are presented in Table 13 and Table 14. Our model (finetuned with our own UltraEdit) can produce more preferable editing results than the baselines, even better than the MagicBrush baseline, which is reported to overfit on its test set.

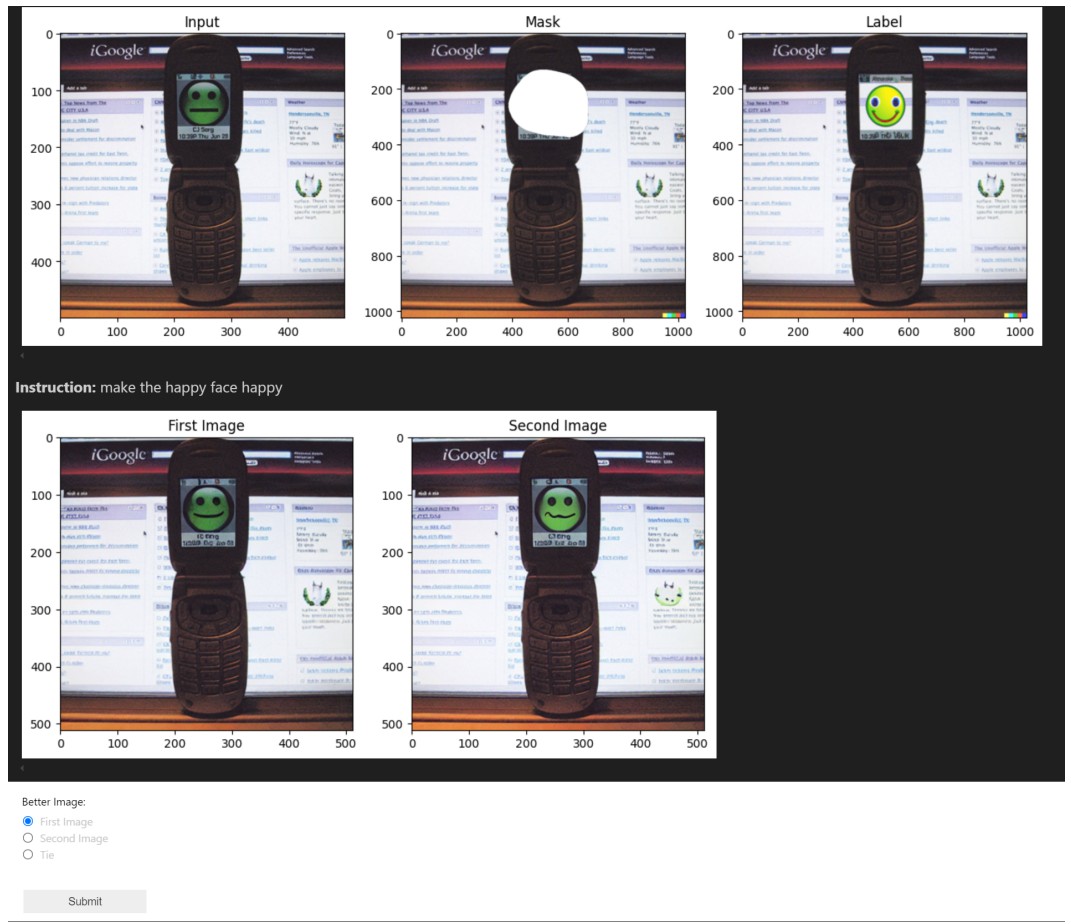

Figure 10: The interface of human evaluation on MagicBrush [71] and Emu test [60] benchmark to evaluate generated images by different models.

|  | SD3 [17] w/ ULTRAEDIT | SDXL [46] w/ ULTRAEDIT | SD1.5 [53] w/ ULTRAEDIT | Emu Edit [60] |
|---|---|---|---|---|
| TrueSkill score | $26.7 \pm 0.7$ | $26.5 \pm 0.7$ | $26.0 \pm 0.7$ | $25.1 \pm 0.7$ |

Table 14: TrueSkill [23] scores of image editing models evaluated by human raters on Emu Test [60] test set.

## D.2 Qualitative Evaluation on Different Benchmarks

In Figure 11 and Figure 12, we present the qualitative examples of different editing tasks on single-turn and multi-turn editing on MagicBrush. In Figure 13, we present the qualitative examples on Emu Edit Test across various editing tasks.

## D.3 Qualitative Evaluation on Real Image Anchors

In Figure 14, we present qualitative results comparing the image editing generation method with and without using real images as anchors. Using real images as anchors to guide the data generation significantly enhances the diversity of the generated images and ensures that the generation results are more stable and aligned with the editing instructions. The image anchors provide substantial information for generation that goes beyond what is conveyed by the image captions alone. Specifically, image anchor ensures visual consistency between the generated source and target images in the image editing pairs, as shown in the first three rows of Figure 14. It can also be observed that with

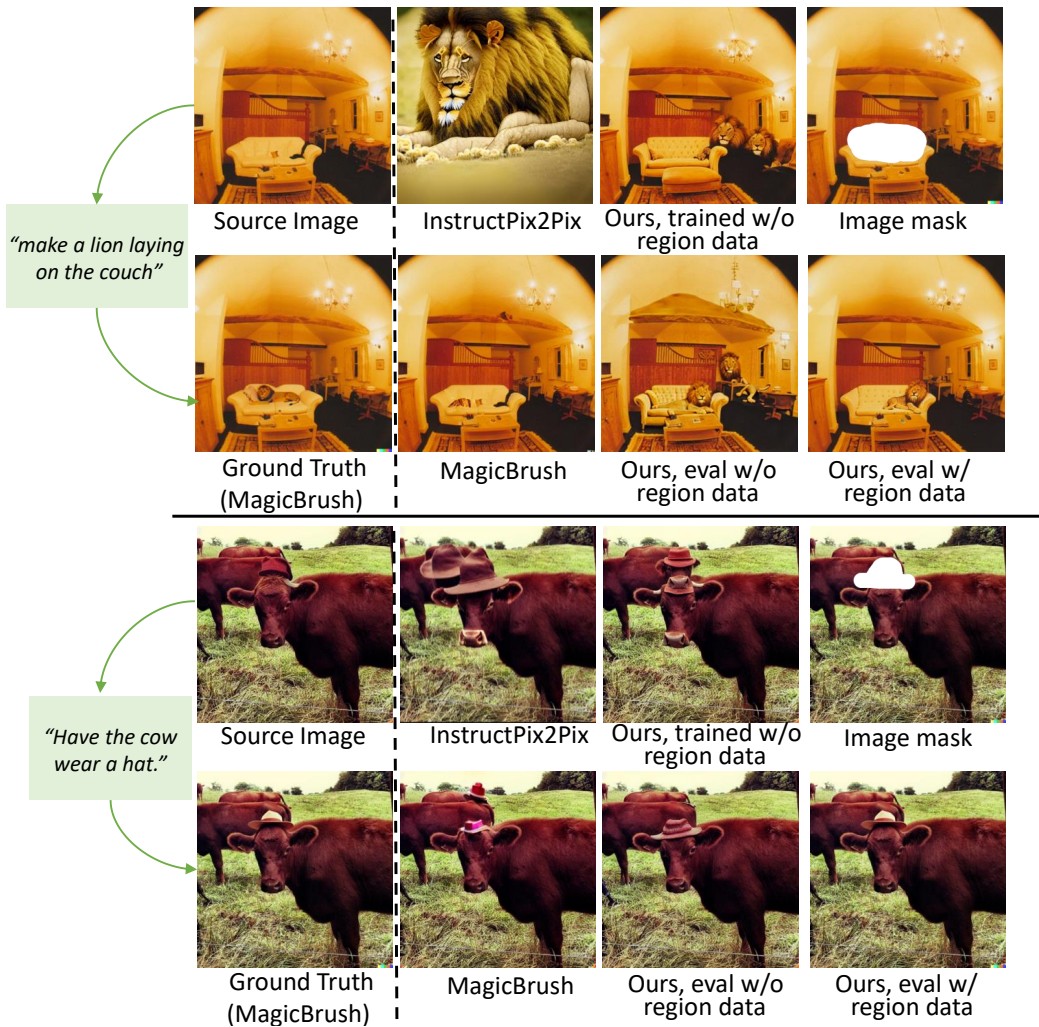

Figure 11: Qualitative evaluation of the model trained on ULTRAEDIT across MagrichBrush benchmark in the Single-turn Setting.

real image anchors, the editing process is more controlled, resulting in fine-grained image edits in the generated samples (see the last three rows in Figure 14).

### D.4 Qualitative Evaluation on Free-form vs. Region-based Editing

In Figure 15, we present qualitative results comparing the model trained with an additional region-based editing task against the model trained solely with free-form image editing data. The comparison highlights that the inclusion of the region-based editing task during training enables the model to perform significantly more precise operations even in the absence of region input during evaluation, especially for background and localized edits.

### D.5 Details of the region-based image editing pipeline

In Stage III, we apply our proposed method for generating region-based images to ensure a seamless transition between inpainted areas and the rest of the image. Initially, we analyze inaccurate masks generated by the segmentation model, as shown in Figure 17. We find these inaccuracies generally fall into a few categories: incorrect identification resulting in overly large masks, masks that are too small for effective editing, fragmented masks from segmentation failures, and fine-grained segment masks that closely resemble the original object, complicating the editing process.

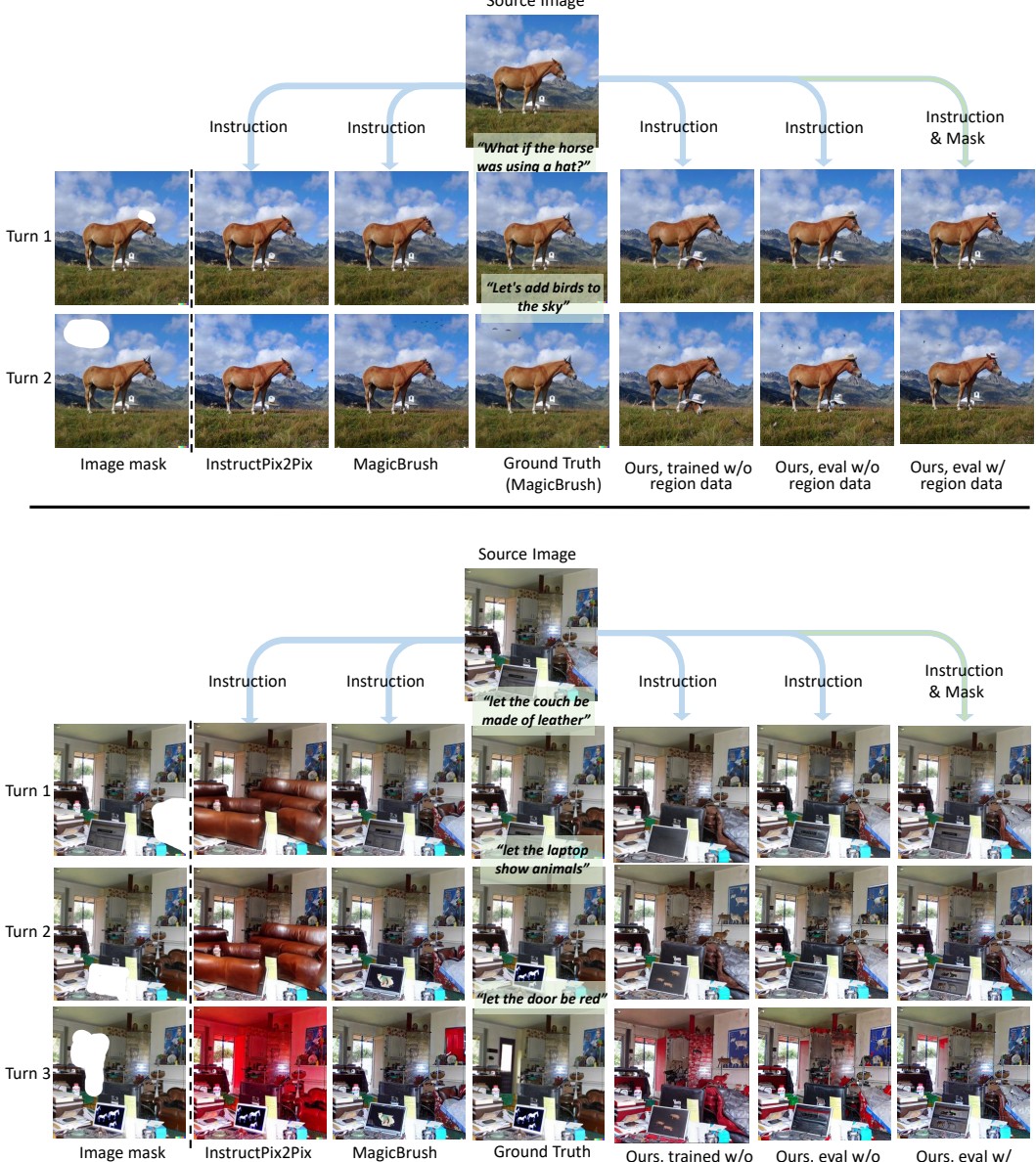

Figure 12: Qualitative evaluation of the model trained on ULTRAEDIT across MagrichBrush benchmark in the Multi-turn Setting.

To address these issues, we filter out excessively large, small, or fragmented masks. Fine-grained masks are adjusted using a soft mask, either a bounding box or contour mask. Our data circulation indicates that our methods significantly reduce artifacts and abrupt boundaries between the mask region and the remaining image. Qualitative evaluations shown in Figure 16 demonstrate the effectiveness of our approach. Images generated without our method exhibit noticeable artifacts along the boundaries of the original and edited regions, highlighting the advantages of our method.

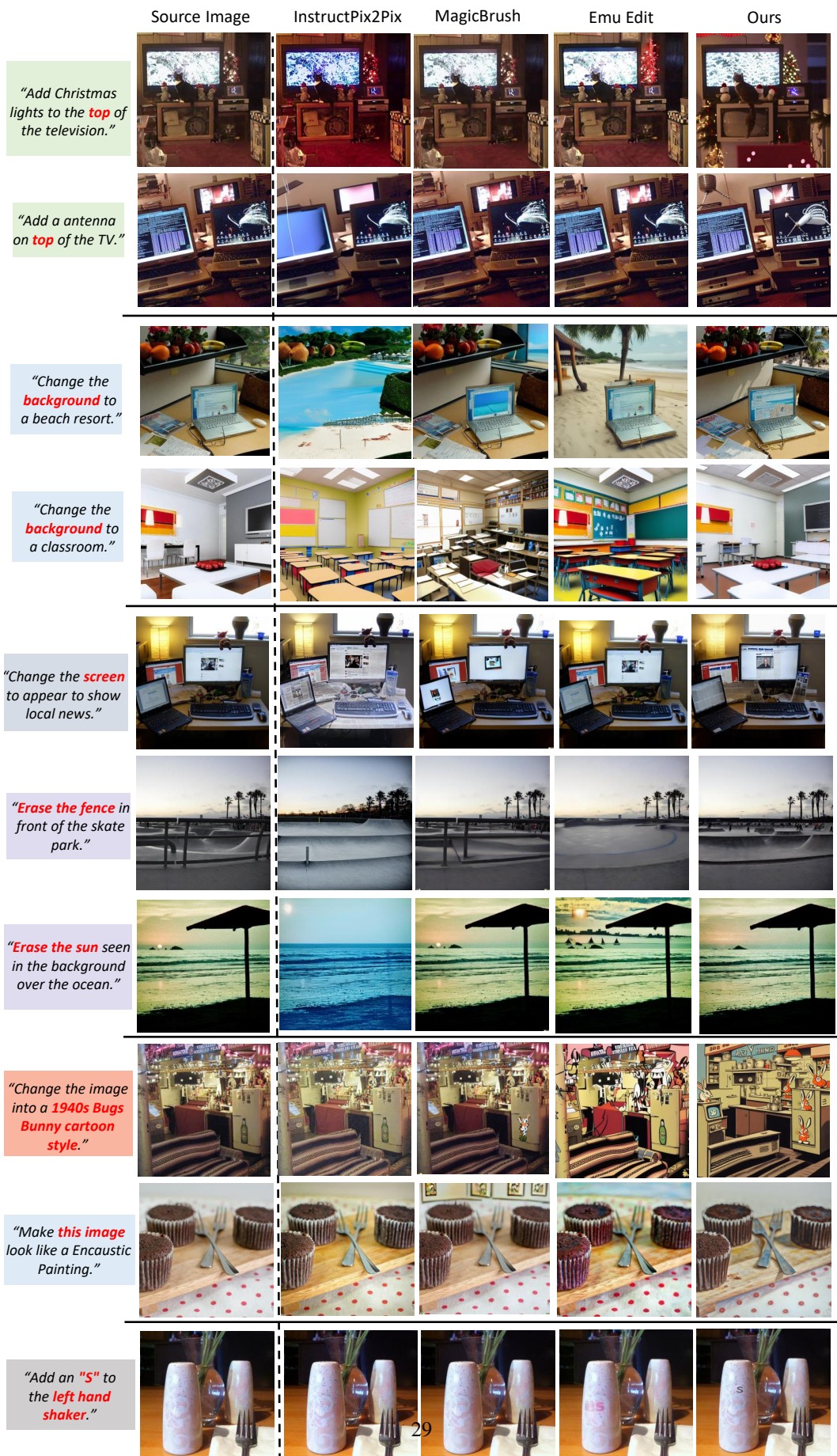

Figure 13: Qualitative evaluation of the model trained with ULTRAEDIT on the Emu Test.

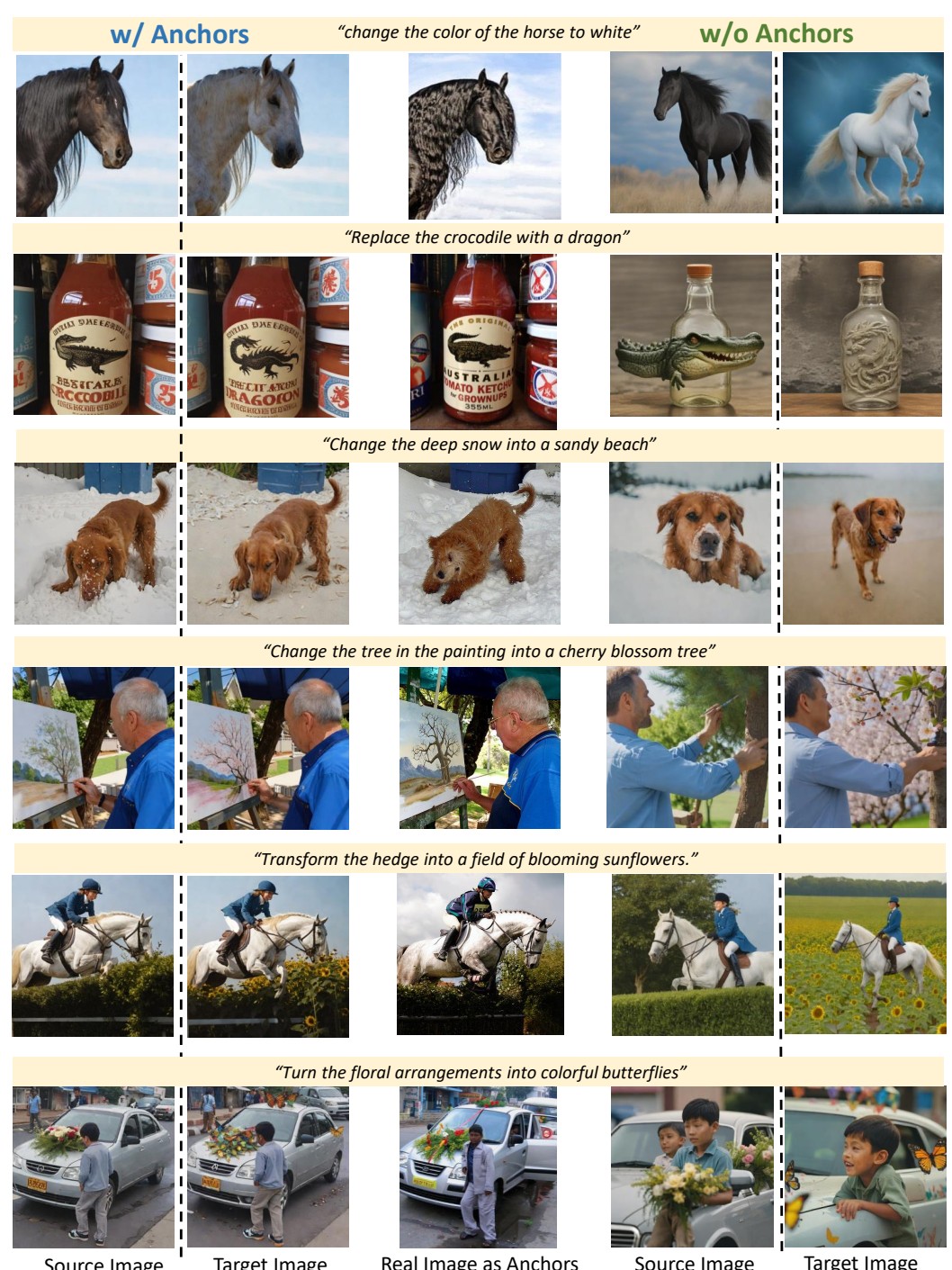

Figure 14: Qualitative evaluation of using real images as anchors during image generation. We compare qualitative examples between the generation pipeline using real image anchors (**left**) and the generation pipeline without real image anchors (**right**). The real images are presented in the middle column.

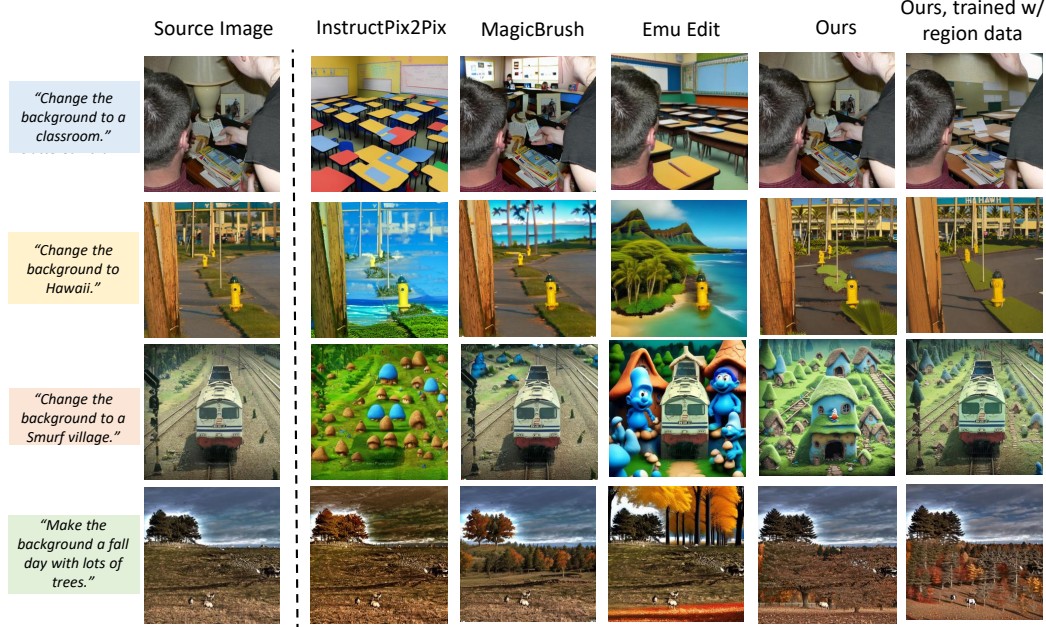

Figure 15: Qualitative evaluation comparing free-form and region-based editing task.

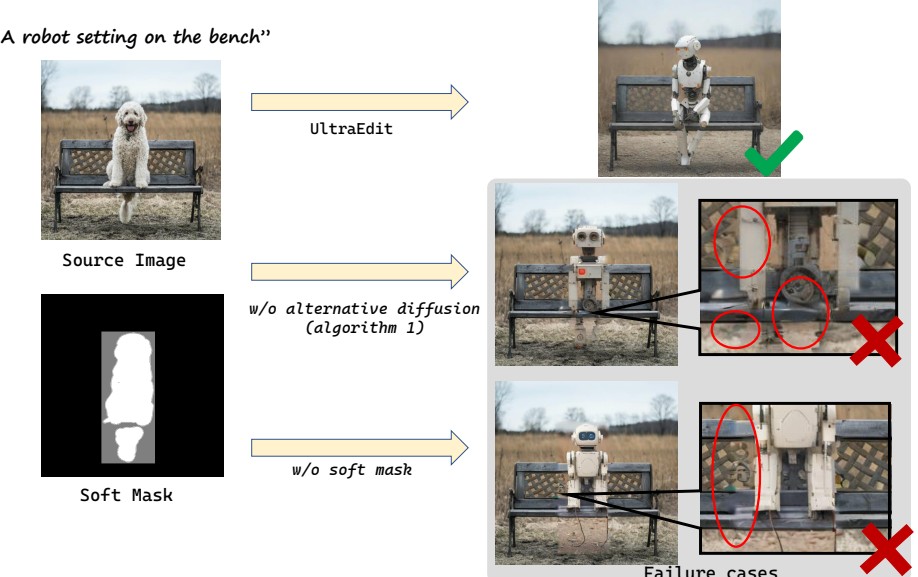

Figure 16: Qualitative evaluations of the region-based image editing pipeline. Generated images Without our method exhibit noticeable artifacts along the boundaries of the original and edited regions, emphasizing pronounced border effects.

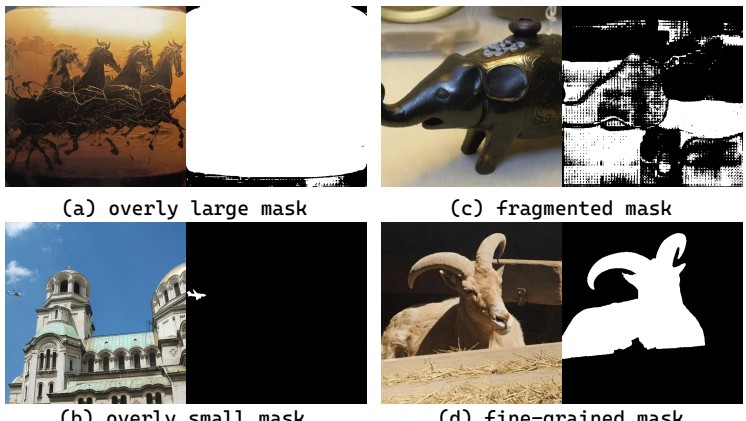

Figure 17: The four main categories of inaccuracies generated masks.

# E  Statement on Limitations and Ethical Concerns

## E.1  Limitations

While ULTRAEDIT represents a significant advancement in instruction-based image editing, several limitations should be acknowledged. Firstly, although our dataset includes diverse editing instructions and real image anchors, the reliance on automatically generated data may introduce some biases and errors. The quality and relevance of the editing instructions, influenced by current large language models and human raters, may not capture all nuances of creative and artistic editing tasks. Furthermore, despite our efforts to provide high-quality region annotations, there may be occasional inaccuracies or inconsistencies in the automatically produced region-based editing data.

Moreover, while our experiments demonstrate the benefits of using real image anchors and region-based editing data, the improvements shown by our diffusion-based editing baselines are benchmark-specific. They may not generalize across all editing scenarios. Future work should focus on enhancing the precision of region annotations and validating the dataset's applicability across a broader range of editing tasks.

Despite these limitations, ULTRAEDIT offers a robust and diverse dataset that significantly contributes to the field of image editing, paving the way for future research and development.

## E.2  Ethical concerns

While UltraEdit offers substantial advancements in the field of instruction-based image editing, several ethical concerns must be considered:

- **Bias and Fairness.** The dataset, while diverse thanks to the efforts on real image anchors, *etc.*, may still contain biases introduced by the automatic generation process and the inherent biases present in the large language models and human raters used. These biases could perpetuate stereotypes or unfair representations in the edited images.

- **Misinformation and Misuse.** The powerful image editing capabilities enabled by UltraEdit could be misused to create misleading or deceptive content, contributing to the spread of misinformation. It is crucial to implement safeguards and promote responsible use of the technology to mitigate this risk.

- **Privacy.** Real image anchors included in the dataset may contain identifiable information. Although efforts have been made to anonymize and protect personal data, there remains a risk of unintentional breaches of privacy.

To address these ethical concerns, we encourage users of UltraEdit to adhere to ethical guidelines, implement robust checks for bias and fairness, and prioritize transparency and accountability in their work. Additionally, we recommend ongoing dialogue within the research community to continuously refine and improve ethical standards in developing and applying image editing technologies.

# F  Datasheet for ULTRAEDIT

We present a Datasheet [22] for documentation and responsible usage of our internet knowledge databases. The required author statement, hosting, licensing, metadata, and maintenance plan can be found in the datasheet.

## F.1  Motivation

**For what purpose was the dataset created?**  We create this large-scale dataset to facilitate research towards image editing based on natural language instructions and regions (masks).

**Who created the dataset (e.g., which team, research group) and on behalf of which entity (e.g., company, institution, organization)?**  This dataset was created by Haozhe Zhao (Peking University), Xiaojian Ma (BIGAI), Liang Chen (Peking University), Shuzheng Si (Tsinghua University), Rujie Wu (Peking University), Kaikai An (Peking University), Peiyu Yu (UCLA), Minjia Zhang (UIUC), Qing Li (BIGAI), and Baobao Chang (Peking University).

## F.2  Distribution

**Will the dataset be distributed to third parties outside of the entity (e.g., company, institution, organization) on behalf of which the dataset was created?**  Yes, the dataset is publicly available on the internet.

**How will the dataset will be distributed (e.g., tarball on website, API, GitHub)?**  All datasets can be downloaded from `https://huggingface.co/`.  Please refer to this table of URL, DOI, and licensing.  The Croissant metadata can be found on the dataset hosting platform (`https://huggingface.co/`).

| Dataset | DOI | License |
|---|---|---|
| ULTRAEDIT-full | 10.57967/hf/2481 | Creative Commons Attribution 4.0 (CC BY 4.0) |
| ULTRAEDIT-free-form-500k | 10.57967/hf/2535 | Creative Commons Attribution 4.0 (CC BY 4.0) |
| ULTRAEDIT-region-based-100k | 10.57967/hf/2534 | Creative Commons Attribution 4.0 (CC BY 4.0) |

**Have any third parties imposed IP-based or other restrictions on the data associated with the instances?**  No.

**Do any export controls or other regulatory restrictions apply to the dataset or to individual instances?**  No.

## F.3  Maintenance

**Who will be supporting/hosting/maintaining the dataset?**  The authors will be supporting, hosting, and maintaining the dataset.

**How can the owner/curator/manager of the dataset be contacted (e.g., email address)?**  Please contact Haozhe Zhao (`mimazhe55360@gmail.com`), Xiaojian Ma (`maxiaojian@bigai.ai`) and Qing Li (`liqing@bigai.ai`).

**Is there an erratum?**  No. We will make announcements if there is any.

**Will the dataset be updated (e.g., to correct labeling errors, add new instances, delete instances)?**  Yes. New updates will be posted on `https://ultra-editing.github.io`.

**If the dataset relates to people, are there applicable limits on the retention of the data associated with the instances (e.g., were the individuals in question told that their data would be retained for a fixed period of time and then deleted)?**  The images in our dataset might contain human subjects, but they are all synthetic.

**Will older versions of the dataset continue to be supported/hosted/maintained?** Yes, old versions will be permanently accessible on huggingface.co.

**If others want to extend/augment/build on/contribute to the dataset, is there a mechanism for them to do so?** Yes, please refer to `https://ultra-editing.github.io`.

### F.4 Composition

**What do the instances that comprise the dataset represent?** Our data is generally stored in the Apache Parquet format, which is a table with multiple columns. We provide images (as source images and target/edited images), captions of source and target images, editing instructions, objects to be edited, metrics (CLIPimg, DINOv2, SSIM, CLIPin, CLIPout, and CLIPdir), and editing regions (optional), as separate columns.

**How many instances are there in total (of each type, if appropriate)?** There are ~4M samples, among which ~100K are region-based editing data, while the rests are free-form editing data.

**Does the dataset contain all possible instances or is it a sample (not necessarily random) of instances from a larger set?** We provide all instances in our Huggingface data repositories.

**Is there a label or target associated with each instance?** No.

**Is any information missing from individual instances?** No.

**Are relationships between individual instances made explicit (e.g., users' movie ratings, social network links)?** No.

**Are there recommended data splits (e.g., training, development/validation, testing)?** No. The entire database is intended for training.

**Are there any errors, sources of noise, or redundancies in the dataset?** Please refer to Appendix E.

**Is the dataset self-contained, or does it link to or otherwise rely on external resources (e.g., websites, tweets, other datasets)?** The dataset is self-contained.

**Does the dataset contain data that might be considered confidential?** No.

**Does the dataset contain data that, if viewed directly, might be offensive, insulting, threatening, or might otherwise cause anxiety?** We have made our best efforts to detoxify the contents via an automated procedure. Please refer to Sec. E.

### F.5 Collection Process

The collection procedure, preprocessing, and cleaning are explained in detail in Section 2 of the main paper.

**Who was involved in the data collection process (e.g., students, crowdworkers, contractors) and how were they compensated (e.g., how much were crowdworkers paid)?** All data collection, curation, and filtering are done by ULTRAEDIT coauthors.

**Over what timeframe was the data collected?** The data was collected between Jan. 2024 and May 2024.

### F.6 Uses

**Has the dataset been used for any tasks already?** Yes, we have used ULTRAEDIT for training our image edit models.

**What (other) tasks could the dataset be used for?** Our dataset is primarily for facilitating research in building more capable image editing models that follow natural language instructions and (optionally) editing region input. Our data might also be used to benchmark existing and future image editing models.

**Is there anything about the composition of the dataset or the way it was collected and preprocessed/cleaned/labeled that might impact future uses?** No.

**Are there tasks for which the dataset should not be used?** We strongly oppose any research that intentionally generates harmful or toxic content using our data.

