# OpenReview forum: "UltraEdit: Instruction-based Fine-Grained Image Editing at Scale"
_NeurIPS.cc/2024/Datasets_and_Benchmarks_Track — NeurIPS 2024 Track Datasets and Benchmarks Poster_

### Official Review · Reviewer_fz2t · 2024-07-21
**Advancing Instruction-based Image Editing with Large Scale Benchmark**

**Rating:** 6
**Confidence:** 4
**Correctness:** Seems good.
**Clarity:** The paper is well written.

**Review:**

The paper is well-written. See ”Strengths" and "Opportunities For Improvement" for pros and cons.

**Strengths:**

- Clear motivation: The paper addresses he drawbacks in existing image editing datasets and highlighting the need for a systematic approach to producing massive and high-quality image editing samples.
- Automated data generation pipeline enabling scalable data: The paper emphasizes the use of an automated data generation pipeline, which allows for the creation of a large-scale dataset with approximately 4 million editing samples.
- Promising experimental results: The experiments conducted using UltraEdit demonstrate its effectiveness. The diffusion-based editing baselines trained on UltraEdit achieve new records on MagicBrush and Emu-Edit benchmarks, indicating improved performance compared to previous methods.

**Additional Feedback:**

Please refer to "Opportunities For Improvement".

**Documentation:**

The dataset and processing pipeline are open-source and well-documented.

**Ethics:**

Seems good. I don't see any major concerns here.

**Limitations:**

Further discussions on reducing noises in the data generation pipeline and biases in MLLMs are encouraged.

**Opportunities For Improvement:**

- One drawback identified in the paper is the absence of support for multi-turn editing. The dataset focuses on instruction-based image editing but does not explicitly address the scenario where multiple editing steps are involved, though experiments show UltraEdit also helps multi-turn editing on MagicBrush.
- The paper is encouraged to delve more into how noise in the dataset, such as inaccurate Grounding DINO and SAM, is mitigated. The presence of noise can impact the quality and reliability of the dataset, potentially affecting the performance of models trained on UltraEdit.
- Instead of directly utilizing real-image editing methods for editing real images, the paper describes the use of a Img2Img diffusion pipeline to generate source images. This indirect approach may introduce additional complexities or limitations in the editing process, potentially affecting the fidelity of the editing results. Why not just use the original real image?
- Bias in LLM-generated editing instructions: The paper does not explicitly address the potential bias present in editing instructions generated by large language models (LLMs). Since LLMs are trained on vast amounts of data, biases present in the training data may inadvertently influence the editing instructions generated. This bias could introduce unintended preferences or limitations in the editing process.

**Relation To Prior Work:**

Seems good.

**Summary And Contributions:**

The paper introduces UltraEdit, a large-scale dataset for instruction-based image editing. UltraEdit overcomes limitations of existing datasets by incorporating the creativity of large language models, using real images for diversity, and supporting region-based editing. Experiments show improved performance on MagicBrush and Emu-Edit benchmarks. UltraEdit sets new standards for diffusion-based editing methods.

---

> ### Author Rebuttal · Authors · 2024-08-17
>
> ### **Q1: the absence of support for multi-turn editing.**
>
> > One drawback identified in the paper is the absence of support for **multi-turn editing**. The dataset focuses on instruction-based image editing but does not explicitly address the scenario where multiple editing steps are involved, though experiments show UltraEdit also helps multi-turn editing on MagicBrush.
> >
>
> ### **A1:**
>
> Thank you for giving me the chance to address the exploration of the multi-turn image editing in our paper. To address the **multi-turn editing scenario**, we have conducted experiments on the MagicBrush benchmark, which includes a set of 1k multi-turn editing cases. These experiments show that our region-based image editing data significantly enhances the model's performance in multi-turn editing tasks. Detailed **results and quality evaluations** for these multi-turn edits are provided in the **Supplemental material.**
>
> We are committed to **incorporating the multi-turn image editing data** into our dataset. Our pipeline is designed to generate instruction-based image editing pairs **across various conditions.** Compare to other methods that either mannaly generated or lack support for automatic multi-step editing, our pipeline is **designed to be modular and flexible.** It allows us to naturally extend to support **generating the multi-step image editing data.** We believe it will further improving the comprehensiveness and utility of our dataset.
>
> ### **Q2: discuss the influence of leading other models into the pipeline**
>
> > The paper is encouraged to delve more into how noise in the dataset, such as inaccurate Grounding DINO and SAM, is mitigated. The presence of noise can impact the quality and reliability of the dataset, potentially affecting the performance of models trained on UltraEdit.
> >
>
> ### **A2:**
>
> To mitigate the noise introduced by **inaccuracies in Grounding DINO and SAM masks**, we employ a comprehensive filtering and correction process:
>
> 1. **Identification of Inaccurate Masks**: We observe that **inaccurate masks** typically fall into the following **categories** (Examples in the **Figure 17 of the general response PDF**):
>     1. Incorrect identification leads to **overly large masks**.
>     2. Masks are **too small** to enable effective editing.
>     3. **Fragmented masks** due to segmentation failures.
>     4. **Fine-grained segment masks** too similar to the original object, complicating the editing process.
> 2. Filtering and Adjustment:
>     1. We **filter out** overly large, small, and fragmented masks. And the fine-grained masks are adjusted **using a soft mask**, which may be either a bounding box mask or a contour mask, to enhance the editing process and make the **transformation of the mask boundary smooth.**
> 3. Quality Assurance: Additionally, we employ **automatic metrics** to evaluate the final generated data's quality. This **end-to-end evaluation** process ensures the dataset's reliability, **minimizing the influence** of inaccuracies from Grounding DINO and SAM.
>
> Thanks for posing this **valid question** and we will **include further details on the noise control mechanisms in the camera-ready** version of the paper.
>
> ### **Q3：Why not just use the original real image**
>
> > Instead of directly utilizing real-image editing methods for editing real images, the paper describes the use of a Img2Img diffusion pipeline to generate source images. This indirect approach may introduce additional complexities or limitations in the editing process, potentially affecting the fidelity of the editing results. Why not just use the original real image?
> >
>
> ### **A3:**
>
> Our approach to using real images as anchors to guide image generation rather than directly editing real images serves two primary purposes:
>
> 1. **Avoidance of Bias from image sources**: Although we gathered real images from diverse sources, using **real images directly could introduce inherent biases** present in the original image dataset, **limiting the generalizability** of our model. For example, only suitable for **real image editing and weak in others**. By using real images as anchors, we ensure a **diverse and controlled dataset**, **minimizing image biases** of the generated dataset.
> 2. **Efficiency**: Editing real images typically involves **time-intensive** processes like DDIM inversion, which requires **hundreds of steps per image.** Our method, on the other hand, can generate **high-quality image editing pairs** in just **1 to 4 steps** using the **real image as anchors**. This significantly reduces the **time and computational resources** needed for dataset generation, allowing for a more efficient and scalable solution.
>
> Our methodology not only ensures a **more unbiased and diverse dataset** but also streamlines the editing process, making it **significantly more efficient without compromising on image fidelity**. We also compare our method with a large range of methods and the Experiment results in **Figure 15 of our general response PDF** confirm that our method performs **competitively with high efficiency** compare with the methods using the real image(namely, **Null-text inversion[1]; Pnp inversion[2]; Brushnet[3]; Powerpaint[4]; MasaCtrl[5]; InfEdit[6]**).
>
> [1] Mokady R, et al. Null-text inversion for editing real images using guided diffusion models[C]//ICCV. 2023: 6038-6047.
>
> [2] Ju X, et al. Pnp inversion: Boosting diffusion-based editing with 3 lines of code[C]//ICLR 2024.
>
> [3] Ju X, et al. Brushnet: A plug-and-play image inpainting model with decomposed dual-branch diffusion[J]. arXiv preprint arXiv:2403.06976, 2024.
>
> [4] Zhuang J, et al. A task is worth one word: Learning with task prompts for high-quality versatile image inpainting[J]. arXiv preprint arXiv:2312.03594, 2023.
>
> [5] Cao M, et al. Masactrl: Tuning-free mutual self-attention control for consistent image synthesis and editing[C]// CVPR 2023: 22560-22570.
>
> [6] Xu S, et al. Inversion-free image editing with natural language[J]. CVPR 2024

---

> > ### Author Rebuttal · Authors · 2024-08-17
> >
> > ## Rebuttal(2/n) by Authors
> >
> > ### **Q4：Bias in LLM-generated editing instructions**
> >
> > > Bias in LLM-generated editing instructions: The paper does not explicitly address the potential bias present in editing instructions generated by large language models (LLMs). Since LLMs are trained on vast amounts of data, biases present in the training data may inadvertently influence the editing instructions generated. This bias could introduce unintended preferences or limitations in the editing process.
> > >
> >
> > ### **A4:**
> >
> > We acknowledge the potential bias in LLM-generated editing instructions and have implemented a comprehensive strategy to address this issue:
> >
> > As detailed in **section 2 and Supplemental material A2**，We incorporate **human-written instructions as in-context learning examples**, addressing the limitations.
> >
> > 1. We first **manually created a few hundred editing instructions** to set a robust baseline. Then, we use these **instructions to guide the language model in generating additional instructions**, followed by a **meticulous manual review** to filter out subpar outputs. The process results in a curated set of **10k high-quality instruction pairs**, detailed in **Table 2 of the supplemental material.**
> > 2. These **10k manually reviewed instruction pairs** are then used as **in-context learning examples** to further drive the language model to **producing instructions with substantial diversity,** ensuring coverage **across a wide range of tasks** compare to other datasets, as shown in **Table 3.**
> >
> > Our pipeline doesn't only **rely on LLMs but combines human intelligence** and machine efficiency to mitigate bias. The curated dataset ensures **coverage across various tasks**, significantly enhancing diversity compared to existing datasets.
> >
> > The future **experimental results** underscore the instruction **diversity and quality** of our pipeline.  The model trained with our datasets shows **superior performance on the Emu Edit Benchmark**, which includes **2000 cases across 8 distinct tasks** and we can see the model **showing superior performance in each task.**

---

> ### Author Response · Authors · 2024-08-20
>
> Dear Reviewer fz2t,
>
> Thank you again for your valuable feedback and comments! We would greatly appreciate it if you could let us know whether you are satisfied with our response. We will be happy to address any remaining concerns.
>
> Sincerely,
>
> Paper991 Authors

---

> ### Author Response · Authors · 2024-08-22
>
> Dear Reviewer fz2t,
>
> I hope this message finds you well. Your detailed review and valuable feedback is greatly appreciated, as it has substantially enhanced the quality of our paper.
>
> Your insightful feedback is immensely valuable to me, and I wish to ensure my responses adequately tackle the problems you have had.
>
> Furthermore, we hope that you will consider a favourable adjustment to your score if our response has mitigated your concerns.
>
> Best,
>
> Paper991 Authors

---

> ### Author Response · Authors · 2024-08-26
>
> Dear Reviewer fz2t,
>
> I hope this message finds you well. We sincerely appreciate your detailed review and valuable feedback, which have significantly enriched the quality of our paper. Your insights are extremely valuable, and we have made every effort to address the issues you raised.
>
> As the rebuttal phase is approaching its conclusion, we hope our responses have alleviated your concerns. We would be grateful if you could consider a favorable adjustment to your score in light of our clarifications.
>
> Best regards,
>
> Authors of Paper991

---

### Official Review · Reviewer_3R87 · 2024-07-25
**UltraEdit: Instruction-based Fine-Grained Image Editing at Scale**

**Rating:** 6
**Confidence:** 4
**Correctness:** Yes
**Clarity:** Yes

**Review:**

Pros:

- the paper proposed a meaningful new dataset of paired instruction-edited images. This will be useful for many research projects along this line.
- the proposed region-based data generation pipeline seems reasonable and can serve as a better starting point for future research in automated instruction data generation.
- the conducted comparions/ablations look comprehensive overall.

Cons:

- directly computing distance (CLIP, SSIM, DINO, etc.) between original and edited images feels suboptimal measurement. We are expecting to see changes between two images, so smaller/larger distance doesn’t mean it’s good or bad. What we need is measurement of similarity in the supposedly invariant portion between the original and edited images? Maybe more meticulous metrics can be proposed to measure such identity preservation.
- to measure the importance of the proposed dataset, I feel it’s better to train the exact same model on the old datasets, proposed dataset, and maybe the combination of all. In this case, we would know how the proposed dataset is indeed helpful. Using different models won’t help us attribute the improvements to the data solely.
- using methods like prompt2prompt or ddim inversion has the issue of identity preservation in details. I feel that the paper misses a section to discuss about the failure cases of the data generation, and how the data curation has been done if any at such large-scale generation. How to guarantee the quality of the constructed pairs is a critical but missing component to me.

**Strengths:**

As mentioned above, the proposed data generation pipeline seems improved over past works, i.e. instructEdit or MagicBrush. The proposed new dataset would benefit research in this area.

**Additional Feedback:**

N/A

**Documentation:**

N/A

**Opportunities For Improvement:**

As mentioned above, I feel we are missing controlled experiments of training same architecture on different datasets, to highlight the importance of the newly proposed data. Also, the paper misses discussion of failure cases in data generation and data curation if any.

**Relation To Prior Work:**

Yes

**Summary And Contributions:**

This paper introduces ULTRAEDIT, a large-scale dataset with approximately 4 million editing samples, designed for instruction-based image editing. It addresses limitations in existing datasets by offering a broader range of editing instructions, using real images for greater diversity, and supporting region-based editing with high-quality annotations. Experiments show that models trained on ULTRAEDIT set new records on benchmarks, and the dataset, along with the code and models, will be made publicly available.

---

> ### Author Rebuttal · Authors · 2024-08-17
>
> ### **Q1: directly computing distance (CLIP, SSIM, DINO, etc.) between original edited images feels suboptimal measurement.**
>
> > directly computing distance (CLIP, SSIM, DINO, etc.) between **original and edited images feels suboptimal measurement**. We are expecting to see **changes between two images**, so smaller/larger distance doesn’t mean it’s good or bad. What we need is measurement of similarity in the supposedly invariant portion between the original and edited images? Maybe more meticulous metrics can be proposed to measure such identity preservation.
> >
>
> ### **A1:**
>
> Thank you for the opportunity to clarify our evaluation approach regarding the image editing pairs' measurements.
>
> As detailed in **section2.3**, We want to clearify that we use the **CLIP-TEXT sim and CLIP-direction** to evaluate the **changes between the images to ensure the instruction following ability** of the generated data.
>
> We employ a **dual strategy** to assess the quality of edited images.
>
> - First, we utilize **CLIP, SSIM, and DINO** to evaluate the similarity at an textual-level, pixel-level and feature-level, ensuring the original and edited images **retain essential common elements**. This is crucial for **identity preservation** in invariant portions.
> - Additionally, we  use **CLIP-TEXT similarity and CLIP-direction** metrics.  CLIP-TEXT similarity calculates the similarity of the **image embedding and the corresponding caption embedding**.  CLIP-direction evaluates similarity of the **changes** in **image embeddin** and the **changes in corresponding caption embedding.**
>
> These ensure that the images have been modified according to given instructions.
>
> By combining these two metric types, we robustly evaluate both the consistency and the directed changes in the images, allowing us to **filter out 95% of suboptimal image pairs**. Detailed statistics and a comparison with other datasets are available in **Table 6 in the Supplemental material**.
>
> This comprehensive metric strategy ensures that we effectively balance between preserving essential image features and achieving desired edits.
>
> ### **Q2： Ensure the comparsion to be fair**
>
> > to measure the importance of the proposed dataset, I feel it’s better to train the **exact same model on the old datasets, proposed dataset**, and maybe the combination of all. In this case, we would know how the proposed dataset is indeed helpful. Using different models won’t help us attribute the improvements to the data solely.
> >
>
> ### **A2:**
>
> Thank you for raising this important point about ensuring **fair comparisons** during the evaluation.
>
> As clarified in **line 145 in Section 3** of our paper, we use the **same diffusion model** architecture (**Stable Diffusion v1.5**) as the backbone for all experiments to **maintain consistency** **with other baselines**.  We also employ **training data volumes identical** to that used in other methods.
>
> To ensure the importance of the proposed dataset, we train our model with the **same amount** of data on the **exact same model** compared with other methods(**450k data trained on Stable Diffusion v1.5**).

---

> > ### Author Rebuttal · Authors · 2024-08-17
> >
> > ## Rebuttal(2/n) by Authors
> >
> > ### **Q3：Data quality and failure case study:**
> >
> > > using methods like prompt2prompt or ddim inversion has the **issue of identity preservation** in details. I feel that the paper misses a section to discuss about the **failure cases** of the data generation, and how the data curation has been done if any at such large-scale generation. How to guarantee the quality of the constructed pairs is a critical but missing component to me.
> > >
> >
> > ### **A3:**
> >
> > Thank you for the chance to clarify the identity preservation issue in our method and data quality assurance of our pipeline.
> >
> > To mitigate the **identity preservation issues**, we use **real images as anchors** to guide image generation. This approach significantly enhances the reliability of identity preservation. As demonstrated in the **ablation study (Table 6)**, using real images as anchors **is a critical factor** in achieving the **high quality dataset.** We also provide the **qualitative evaluation of the failure case analysis in figure 8 in Supplemental material**, comparing the images generated **with or without the image anchors.**
> >
> > Moreover, we have made further discussion about the **inaccurate masks** generated by segment model (illustrated in **Figure 17 of the attached PDF**), and we have provided an **ablation study of the region-based image editing pipeline** in **Figure 16 of the attached PDF**. It reveals that images generated **without our method exhibit noticeable artifacts** along the boundaries of original and edited regions, highlighting pronounced border effects. We are **committed** to including **these discussions in the camera-ready version** of the paper.
> >
> > To ensure high data quality, we employ a **comprehensive automatic metric** system as a safeguard during image generation to ensure high data quality, successfully **filtering out 95% of suboptimal data.**
> >
> > To further validate this automated process, we performed a **human evaluation** on a random subset of 200 image pairs. The results indicated a strong alignment between our automatic metrics and human assessments, affirming the reliability of our filtering approach.  Below is a summary of the evaluation:
> >
> > |  | Success Cases | Failed Cases | Metric filtered |
> > | --- | --- | --- | --- |
> > | With Safeguard | 188 | 12 | \ |
> > | Without Safeguard | 16 | 184 | 192 |
> >
> > **Experiments, together with further human evaluations** detailed in **Table4 and Table 7 (in Supplemental material)**, also demonstrate the **superior quality** of UltraEdit data compared to other datasets.
> >
> > |  | Ours(SD1.5) | MagicBrush | InstructPix2Pix |
> > | --- | --- | --- | --- |
> > | TrueSkill score | 25.5 ± 0.8 | 23.7 ± 0.7 | 22.6 ± 0.7 |
> >
> > We also evaluate the performance of **various model backbones** trained on our dataset using human evaluation.
> >
> > |  | SD3 w/ UltraEdit | SDXL w/ UltraEdit | SD1.5 w/ UltraEdit | Emu Edit |
> > | --- | --- | --- | --- | --- |
> > | TrueSkill score | 26.7 ± 0.7 | 26.5 ± 0.7 | 26.0 ± 0.7 | 25.1 ± 0.7 |
> >
> > ### **Q4：missing controlled experiments**
> >
> > > As mentioned above, I feel we are missing controlled experiments of **training same architecture on different datasets**, to highlight the importance of the newly proposed data. Also, the paper misses discussion of failure cases in data generation and data curation if any.
> > >
> >
> > ### **A4:**
> >
> > Thank you for valuable feedback.
> >
> > Regarding controlled experiments, as clarified in line 145 in Section 3, we ensured a fair comparison by using the **same architecture**, specifically Stable Diffusion v1.5, trained **across different datasets, including InstructPix2Pix, MagicBrush, and Hive**. These experiments were conducted on the model trained with the **same amount of data of different datasets** to demonstrate the **impact and importance** of our proposed UltraEdit data.
> >
> > Also, we will include **more failure cases** during the data generation in the camera-ready paper. We will provide a **comprehensive analysis and commit to including more detailed discussions** of these cases.
> >
> >
> > We trust this clarification addresses your concerns, and we hope you can kindly increase your score if our response was helpful.

---

> ### Author Response · Authors · 2024-08-20
>
> Dear Reviewer 3R87,
>
> Thank you again for your valuable feedback and comments! We would greatly appreciate it if you could let us know whether you are satisfied with our response. We will be happy to address any remaining concerns.
>
> Sincerely,
>
> Paper991 Authors

---

> ### Author Response · Authors · 2024-08-22
>
> Dear Reviewer 3R87,
>
> We sincerely appreciate your careful review and valuable feedback, which has substantially improved our paper.
>
> Your feedback is highly valuable to me, and I want to make sure my responses effectively address the issues you have had. If you have any further questions or require additional clarification, please feel free to reach out.
>
> We also hope that you can kindly increase your score if our response has helped address your concerns.
>
> Best,
>
> Paper991 Authors

---

> ### Author Response · Authors · 2024-08-26
>
> Dear Reviewer 3R87,
>
> **We would like to express my sincere gratitude for the time and effort you have dedicated to evaluating our submission.**
>
> As the rebuttal is nearing its conclusion, if you have any further questions or require additional clarification, please do not hesitate to reach out to us. We sincerely hope that our responses satisfactorily resolve your queries and that you might consider revising your score in light of the clarifications provided.
>
> Thank you once again for your thoughtful review.
>
> Sincerely,
>
> Authors of Paper991

---

> ### Author Rebuttal · Authors · 2024-08-31
>
> Dear Reviewer 3R87,
>
> Thank you for your thoughtful review of our manuscript. We've carefully addressed your comments in our rebuttal and made revisions to improve the work.
>
> As the discussion period will end in 1 day, we kindly request that you review our rebuttals. We would like to know whether they address your concerns or if there are any further insights you might have. Your expertise is greatly appreciated.
>
> Sincerely,
>
> Authors of Paper991

---

### Official Review · Reviewer_j7cR · 2024-07-30
**emergency review**

**Rating:** 6
**Confidence:** 2
**Correctness:** as far as I can tell, the paper is co…
**Clarity:** the paper is well-written.

**Review:**

This is an emergency review. I like this work and the contribution is relevant to the community. The dataset and models help improve image editing with generative models. Yet there are quality issues regarding diversity.

**Strengths:**

- large dataset and models
- performance looks good
- I like the categorization of the edit instructions.
- the multi turn results look promising

**Additional Feedback:**

this is an emergency review

**Documentation:**

I appreciate the detailed discussion of limitations, the ethical statement, and the dataset documentation in the supplement. However, these parts (not entirely but the central parts) have to be moved to the main paper.

**Ethics:**

I appreciate the detailed discussion of limitations, the ethical statement, and the dataset documentation in the supplement. However, these parts (not entirely but the central parts) have to be moved to the main paper. Please also check your generated dataset for unsafe content with proposed methods (LlavaGuard).

**Limitations:**

I appreciate the detailed discussion of limitations, the ethical statement, and the dataset documentation in the supplement. However, these parts (not entirely but the central parts) have to be moved to the main paper.

**Opportunities For Improvement:**

- I'm unsure about the actual diversity of the prompts. using large models has shown to have limited diversity. But there are methods to actually ensure diversity such as "quality diversity" (https://arxiv.org/abs/2310.12103).
- for evaluation, I encourage the authors to use more robust image similarity methods than L1 and L2 such as LPIPS
- how do the authors ensure quality in categorizing the edit instructions?
- how do the authors ensure safety in their dataset? e.g. take a look at schramowski et al. safe latent diffusion for safety during generation, or at Helff et al. Llavaguard for safety after generation.

**Relation To Prior Work:**

good reference to related work. There is very relevant related work that also provides image editing methods and datasets:

- Brack et al. LEdits++ at CVPR

**Summary And Contributions:**

This paper proposes a novel dataset to train and evaluate image manipulation methods. Their primary contribution is a larger and more diverse set. At the same time, they provide the pipeline to achieve the dataset and models trained and evaluated on this set.

---

> ### Author Rebuttal · Authors · 2024-08-17
>
> ### **Q1: Instruction diversity and quality.**
>
> > I'm unsure about the actual diversity of the prompts. using large models has shown to have limited diversity. But there are methods to actually ensure diversity such as "quality diversity" (https://arxiv.org/abs/2310.12103).
> >
>
> ### **A1:**
>
> To ensure diversity and quality in instruction generation, we incorporate **human-written instructions as in-context learning examples**, addressing the limitations observed in large language models. Our process involves:
>
> 1. We first **manually created a few hundred editing instructions** to set a robust baseline. Then, we use these instructions to guide the language model in **generating additional instructions**, followed by a **meticulous manual review** to filter out subpar outputs. The process results in a curated set of **10k high-quality instruction pairs,** detailed in **Table 2 of the supplemental material.**
> 2. These **10k manually reviewed instruction pairs** are then used as **in-context learning examples** to further drive the language model to producing **instructions with substantial diversity**, ensuring coverage **across a wide range of tasks** compare to other datasets, as shown in **Table 3.**
>
> The **further experimenta**l results underscore the **instruction diversity and quality of our pipeline**.  The model trained with our datasets shows **superior performance** on the **Emu Edit Benchmark**, which includes **2000 cases across 8 distinct tasks** and we can see the model **showing superior performance in each task.**
>
> ### **Q2: Evaluation problem.**
>
> > for evaluation, I encourage the authors to use more robust image similarity methods than L1 and L2 such as LPIPS.
> >
>
> ### **A2:**
>
> Thank you for your suggestion regarding the evaluation of image editing performance.
>
> In our paper, we **adhere** to the **evaluation metrics** established by **prior works** [1,2] to ensure a **fair comparison**. Specifically:
>
> - MagicBrush Benchmark: Evaluates the similarity of **generated images to ground truth**.
> - Emu Edit Benchmark: Focuses on the **CLIP similarity between target captions and generated images** as it **does not provide** ground truth images.
>
> Also, to ensure the robustness of the evaluation metric, during the **dataset generation,** we also use the **SSIM, CLIP and DINO similarity** to ensure the **pixel-level, textual-level and feature-level similarity** between the source and target images.
>
> [1] Zhang, Kai, et al. "Magicbrush: A manually annotated dataset for instruction-guided image editing." *Advances in Neural Information Processing Systems* 36 (2024).
>
> [2] Sheynin, Shelly, et al. "Emu edit: Precise image editing via recognition and generation tasks." *Proceedings of the IEEE/CVF Conference on Computer Vision and Pattern Recognition*. 2024.
>
> We acknowledge the **robustness of LPIPS** as a similarity measurement and agree that it can enrich our evaluation process. We **commit to integrating LPIPS** into our evaluation pipeline to provide a more comprehensive assessment of image similarity.
>
> By including these diverse evaluation metrics, we ensure a thorough and multifaceted evaluation of the image editing performance of models trained with our dataset.
>
> ### **Q3: Quality in categorizing the edit instruction.**
>
> > how do the authors ensure quality in categorizing the edit instructions?.
> >
>
> ### **A3:**
>
> Ensuring the quality of categorizing edit instructions is an important task in our methodology. We implement a two-step approach to maintain efficiency and high standards:
>
> 1. **Automated Categorization:** We utilize GPT-3.5 to categorize the edit instructions efficiently and consistently.
>
> 2. **Human Evaluation:** To ensure the accuracy and reliability of the automated categorization, we conduct a thorough human evaluation on a randomly selected subset of the categorized instructions.
>
> ### **Q4: safety in the dataset**
>
> > how do the authors ensure safety in their dataset? e.g. take a look at schramowski et al. safe latent diffusion for safety during generation, or at Helff et al. Llavaguard for safety after generation..
> >
>
> ### **A4:**
>
> Thank you for raising this issue.
>
> 1. To ensure the safety of our dataset, following others, we also implement a **safety checker[4]** based on the CLIP model, which has been pretrained on harmful content. This checker evaluates the safety of images during generation.
> 2. Additionally, we conducted a **preliminary human evaluation on a subset** of the dataset to further ensure safety.
> 3. In response to your suggestions, we have **assessed the effectiveness of safety tools** like Llavaguard on our dataset. The results demonstrated its utility in enhancing image safety post-generation. We will **integrate Llavaguard into our dataset curation process** to provide an additional layer of safety validation.
>
> We are committed to ensuring the **highest safety standards** throughout our data generation and curation processes, and we **appreciate your guidance on this matter**. We also see this as a **promising direction** for my future research.
>
> [4] Rombach, Robin, et al. "High-resolution image synthesis with latent diffusion models." *Proceedings of the IEEE/CVF conference on computer vision and pattern recognition*. 2022.

---

> > ### Author Rebuttal · Authors · 2024-08-17
> >
> > ## Rebuttal(2/n) by Authors
> >
> > ### **Q5: Add detailed discussion of limitations**
> >
> > > I appreciate the detailed discussion of limitations, the ethical statement, and the dataset documentation in the supplement. However, these parts (not entirely but the central parts) have to be moved to the main paper.
> > >
> >
> > ### **A5:**
> >
> > Thank you for your valuable feedback. We agree that the detailed discussion of limitations, the ethical statement, and the dataset documentation are crucial elements that deserve prominence in the main paper. We will **ensure that these essential sections are integrated into the main paper** in the camera-ready version.
> >
> > ### **Q6: Add LEdits++ FOR references**
> >
> > > good reference to related work. There is very relevant related work that also provides image editing methods and datasets:Brack et al. LEdits++ at CVPR
> > >
> >
> > ### **A6:**
> >
> > Thank you for highlighting LEdits++ as a relevant related work. It is surely an insightful work and can provide a well-rounded context and reinforce the soundness of our work.
> >
> > We will **ensure that LEdits++ is included in our pape**r. This addition will not only add depth to our literature review but also provide valuable insights and a basis for comparison to our methodology.
> >
> > We trust that this additional information has addresses your concerns. If you have any further **questions or require additional details**, **please do not hesitate to reach out**.
> >
> > We also hope that you can kindly increase your score if our response has helped address your concerns.

---

> > ### Comment · Reviewer_j7cR · 2024-08-19
> >
> > Thanks for your response and commitment. Can you please provide more details about the statements made? For example, you talk about human evaluations on instruction categorization and dataset safety.

---

> > > ### Author Response · Authors · 2024-08-20
> > >
> > > Dear Reviewer j7cR,
> > >
> > > Thank you again for your valuable feedback and comments! We would greatly appreciate it if you could let us know whether you are satisfied with our response. We will be happy to address any remaining concerns.
> > >
> > > Sincerely,
> > >
> > > Paper991 Authors

---

> ### Author Rebuttal · Authors · 2024-08-19
>
> Thank you for seeking further clarification. Here’s a detailed breakdown of our processes for instruction categorization and dataset safety ensurement:
>
> **Instruction Categorization:**
>
> 1. **Visualization and Categorization:**
>     - Following the **previous works** [2,3],  we **initiate our categorization** by **visualizing** the distribution of instruction types and keywords based **on word frequency.**
>     - Using these visualizations as a guideline, we **identified and established nine distinct categories** for our instructions. We leverage GPT-3.5 to **classify these instructions** by providing it with the original captions, target captions, and edit instructions through a structured prompt:
>
>         ---
>
>         The following prompt provides an instruction for image editing, along with the original and revised image captions, and a set of instruction categories.
>
>         **Edit Instruction:** "{edit_instruction}"
>
>         **Original Image Caption:** "{input_text}"
>
>         **Revised Image Caption:** "{output_text}"
>
>         **Instruction Categories:** {object_list}
>
>         Your task is to determine which categories best fit this example, based on the information above. Please use precise categories.
>
>         **Note:**
>
>         1. If you cannot identify a category, please respond with "Others."
>         2. Your response should solely identify the category of the instruction type, without any additional context or details.
>
>         **Response:**
>
>         ---
>
> 2. **Validation via Human Evaluation:**
>     - To evaluate the accuracy of GPT-3.5’s categorization, we engaged **three highly-educated individuals, each pursuing advanced degrees** (Master's or Ph.D.), to assess the quality of the categorizations. These evaluators were provided with **500 randomly selected** instruction pairs to assess the alignment of GPT's categorization with human judgment.
>     - For each evaluation, the evaluators were given:
>         - Original Image Caption
>         - Revised Image Caption
>         - Edit Instruction
>         - GPT-3.5 Assigned Category
>
>         Evaluators were directed to **review each instruction pair against nine predefined categories**. They assessed whether GPT-3.5’s categorizations were accurate or whether another category might better fit the context provided by the captions and instructions. Evaluators should mark categorizations as **"Correct,"  or "Incorrect,"** and offer justification for any deviations from GPT-3.5’s categorization.
>
>
> **Dataset Safety:**
>
> 1. **Generation Pipeline Safety Measures:**
>     - Following **perivous works**[1,4,5,6], we employ a **safety checker** provied by the Hugging Face[4] and CompVis[1], which uses a ‘NSFW’ filter during the generation process. This filter employed a pretrained CLIP embeddings to detect harmful content. The checker checking compars the *generated*  image CLIP  embedding against known hard-coded NSFW concepts to detect the harmful content.
> 2. **Human Evaluation in Instruction Generation:**
>     - To ensure the safety of editing instructions, **eight highly-educated individuals**, as detailed above, carefully **annotated and reviewed** the generated **instructions** to ensure they are **safe** and devoid of harmful content. These manually vetted high-quality instructions were used as **in-context examples** to guide further large-scale instruction generation by the language models.
> 3. **Post-Generation Assessment:**
>     - After image generation, the same team of **eight evaluators** conduct a comprehensive review of the generated samples, checking both the source and target images for harmful content. Each evaluator reviewed **100 randomly selected image pairs**, ensuring the absence of unsafe material. Evaluators were asked to consider both the source and target images within each pair and  categorized content as **"Safe" or"Unsafe”.**
>     - Evaluators were provided with detailed instructions outlining the criteria for assessing image safety. The evaluation focused on identifying:
>         - Explicit or sexual content.
>         - Harassment, harmful, or violent imagery.
>         - Any other content that could cause discomfort to the evaluators.
>
> We will also integrate advanced automated tools such as Llavaguard to augment our safety verification processes. And We remain committed to addressing any subsequent safety concerns by implementing timely and effective measures to safeguard our data.
>
> We appreciate your attention to these critical quality and safety issues and welcome any further suggestions that can aid in enhancing our processes.
>
> [1] Rombach, Robin, et al. "High-resolution image synthesis with latent diffusion models." Proceedings of the IEEE/CVF conference on computer vision and pattern recognition. 2022.
>
> [2] Hui, Mude, et al. "Hq-edit: A high-quality dataset for instruction-based image editing." *arXiv preprint arXiv:2404.09990* (2024).
>
> [3] Zhang, Kai, et al. "Magicbrush: A manually annotated dataset for instruction-guided image editing." *Advances in Neural Information Processing Systems* 36 (2024).
>
> [4] von Platen, et al.. “Diffusers: State-of-the-art diffusion models. https://github.com/huggingface/diffusers (2022)”
>
> [5] Liu, Runtao, et al. "Latent guard: a safety framework for text-to-image generation." *arXiv preprint arXiv:2404.08031* (2024).
>
> [6] Patil, Suraj, et al. "amused: An open muse reproduction." *arXiv preprint arXiv:2401.01808* (2024).

---

> ### Comment · Reviewer_j7cR · 2024-08-20
>
> thanks for the detailed response but what did the human evaluation actually look like? did gpt3.5 perform perfectly on this task or only 70% accurate? same for the safety eval.
>
> What are the results of applying LlavaGuard to your dataset?

---

> > ### Author Rebuttal · Authors · 2024-08-20
> >
> > Thank you for your continued engagement!
> >
> > We have included screenshots of the human evaluation interface for both the safety and categorization checks; these are found in Figures 17 and 18 of the attached PDF.
> >
> > Regarding the safety evaluation, all reviewers agreed that the sampled subset of our dataset is free from safety issues.
> > Moreover, we utilized high-quality real images sourced from open-source datasets as guidance for image generation in our pipeline. As a result, our dataset images inherently adopt the same safety standards as those established by these open-source datasets.
> >
> > Additionally, for the evaluation of the LLavaGuard, we evaluated 50 sample images from our dataset using the demo provided by the LLavaGuard and all samples reveived  a "rating": "Safe". We highly commend LLavaGuard as it offers a seamless way to automate image safety evaluation. Integrating it into our pipeline enhances our dataset's safety without additional effort.
> >
> > Regarding the automatic categorization quality, GPT-3.5 performed exceptionally well in this sample task. Our human evaluation indicated that GPT-3.5 achieved over 95% accuracy compared to human evaluators, (1451 out of 1500 cases.)
> >
> > We are committed to maintaining high standards across our data processes and appreciate your guidance on further improving our methodology.
> > Your feedback is highly valuable to me, and I want to make sure my responses effectively address the issues you have had. If you have any further questions or require additional clarification, please feel free to reach out.

---

> > > ### Comment · Reviewer_j7cR · 2024-08-22
> > >
> > > Thanks for the clarifications. I cannot access uploaded files during the rebuttal as detailed by the NeurIPS committee.
> > > I'm unsure about the results provided without further evidence but appreciate the author's effort. Hence I keep my score to accept this paper with a borderline accept score.

---

> > > > ### Author Response · Authors · 2024-08-22
> > > >
> > > > Dear Reviewer j7cR,
> > > >
> > > > Thank you for your continued feedback and for recognizing our efforts.
> > > >
> > > > To address your concerns, we are pleased to inform you that our entire dataset is open-sourced and well-documented.  The new results and evaluations present in our discussion attachments will be made available on our project page.
> > > >
> > > > During the dataset creation and categorization, we follows the pervious works and we adhered the same safety standards of the open-sourced image caption datasets used in our data creation.
> > > >
> > > > Regarding the PDF upload, OpenReview allows for a one-page PDF during the rebuttal period, containing only figures and tables. We will clarify the accessibility of these files with the Area Chair for clarification on the accessibility of uploaded files.
> > > >
> > > > We appreciate your engagement and consideration of our work.
> > > >
> > > > Best,
> > > >
> > > > Paper991 Authors

---

> > ### Author Response · Authors · 2024-08-22
> >
> > Dear Reviewer j7cR,
> >
> > We sincerely appreciate your careful review and valuable feedback, which has substantially improved our paper.
> >
> > Your feedback is highly valuable to me, and I want to make sure my responses effectively address the issues you have had. If you have any further questions or require additional clarification, please feel free to reach out.
> >
> > We also hope that you can kindly increase your score if our response has helped address your concerns.
> >
> > Best,
> >
> > Paper991 Authors

---

### Official Review · Reviewer_RFxS · 2024-07-30
**This review evaluates the work of ULTRAEDIT. Through a thorough reading and examination of the paper, the evaluation covers its strengths and weaknesses, areas for improvement, credibility, correctness, and clarity. The reviewer recommends accepting this work.**

**Rating:** 7
**Confidence:** 4

**Review:**

Quality:
1.The dataset contains over 4 million high-quality editing examples, far exceeding prior datasets. Metrics show edits accurately follow instructions while preserving source image quality.
2.To validate the effectiveness of the proposed dataset, the authors employed multiple existing evaluation methods to further verify its quality. Rich experiments were conducted, including ablative studies, to comprehensively test the dataset. The experimental settings covered various scales and types of data from ULTRAEDIT. Furthermore, the experimental results demonstrated the dataset's capability of pushing the state-of-the-art, as diffusion models trained on ULTRAEDIT achieved new records on challenging benchmarks like MagicBrush and Emu Edit Test. This rigorous experimental validation with ablations firmly established the dataset's utility in advancing the field.
Clarity:
1.The paper clearly motivates the work by analyzing limitations of prior datasets and approaches.
2.The methodology to generate ULTRAEDIT through leveraging language models, real images and region data is clearly described.
3.Experimental setup, results and analyses are clearly reported to validate the effectiveness of the proposed dataset.
Originality:
1.Prior datasets and approaches in the literature exhibited certain limitations. ULTRAEDIT was proposed and designed deliberately to address the shortcomings identified in previous works. Specifically, existing datasets generally relied solely on text-to-image generation models for data synthesis, which restricted instruction diversity and introduced biases to the data distribution. Meanwhile, prior methods lacked region-based editing support, limiting their capabilities. Distinct from these, ULTRAEDIT put forth novel techniques that leveraged large language models, real-image anchors as well as automatic region extraction. Such original techniques served to mitigate the biases, enhance instruction variety, and enable region-guided editing - all of which had been missing from prior art. Therefore, through critical analysis of the field and the design of creative solutions, ULTRAEDIT advanced the technology in an inventive manner and made an original contribution to the domain.
2.ULTRAEDIT is the largest instruction-based image editing dataset to be released to the public.
Significance:
1.ULTRAEDIT addresses key limitations of prior work to advance the field of instruction-based image editing.
2.Models trained on it achieve SOTA results on benchmarks, confirming its role in pushing the technology forward.
3.As a large, high-quality dataset, it will serve as an important resource to benefit future research through public availability.

**Strengths:**

The strengths of this study are as follows: It addresses the major limitations of previous datasets with ULTRAEDIT, significantly advancing instruction-based image editing. The public dataset it provides will serve as a valuable resource for future research. ULTRAEDIT is highly relevant to the fields of computer vision and image editing research, making significant contributions to studies in these areas.

**Additional Feedback:**

NO

**Clarity:**

The paper is well-written, with precise language, clear illustrations, and a well-organized format. It clearly motivates the work by analyzing limitations of prior datasets and approaches. The methodology to generate ULTRAEDIT through leveraging language models, real images and region data is clearly described. Experimental setup, results and analyses are clearly reported to validate the effectiveness of the proposed dataset.

**Correctness:**

Based on the provided materials, the claims made in the submission appear to be correct, and the dataset seems to be constructed in a sound way. The authors have addressed key aspects such as instruction diversity, real image anchors, and region-based editing. However, a full evaluation would require a thorough review of the entire document and dataset.

**Documentation:**

Yes, the submission provides sufficient detail on data collection and organization, availability and maintenance, and ethical and responsible use. It includes documentation and intended uses, a URL for reviewer access to the dataset, and a hosting, licensing, and maintenance plan.

**Limitations:**

Based on the provided excerpts, the claims made in the submission appear to be correct, and the dataset seems to be constructed in a sound way. The authors have addressed key aspects such as instruction diversity, real image anchors, and region-based editing.

**Opportunities For Improvement:**

Although ULTRAEDIT marks significant progress in the field of instruction-based image editing, several limitations remain. Firstly, while the dataset includes diverse editing instructions and real image anchors, the reliance on automatically generated data may introduce certain biases and errors. The quality and relevance of the editing instructions, influenced by current large language models and human raters, may not fully capture all the nuances of creative and artistic editing tasks. Additionally, despite the team's efforts to provide high-quality region annotations, the automatically generated region-based editing data may still occasionally exhibit inaccuracies or inconsistencies. Furthermore, although experiments demonstrate the advantages of using real image anchors and region-based editing data, the improvements shown by the diffusion-based editing baselines are specific to these benchmarks and may not generalize across all editing scenarios. Future work should focus on enhancing the precision of region annotations and validating the dataset's applicability across a broader range of editing tasks.

**Relation To Prior Work:**

Yes, it considers the differences from previous datasets and works, and improves upon them by analyzing their shortcomings.

**Summary And Contributions:**

This work propose an innovative automatic pipeline to generate image editing data that addresses problems found in existing datasets, such as limited instruction diversity, image biases, and the lack of region-based editing data. Using this pipeline, the authors create ULTRAEDIT, a large-scale, high-quality image editing dataset featuring diverse instructions, real images as anchors, and additional editing region annotations. They perform extensive studies to demonstrate how a canonical diffusion-based image editing model benefits from ULTRAEDIT, offering insights and analysis on key design principles.

---

> ### Author Rebuttal · Authors · 2024-08-17
>
> ### **Q1: the reliance on automatically generated data may introduce certain biases and errors.**
>
> > Although ULTRAEDIT marks significant progress in the field of instruction-based image editing, several limitations remain. Firstly, while the dataset includes diverse editing instructions and real image anchors, the reliance on automatically generated data may introduce certain biases and errors. The quality and relevance of the editing instructions, influenced by current large language models and human raters, may not fully capture all the nuances of creative and artistic editing tasks.
> >
>
> ### **A1:**
>
> Thank you for noticing the error prevention mechanisms in our method. As **elucidated in Section 2.3** of our manuscript, we employ **an automatic metric** for data filtering, which acts as a **safeguard during data generation**.
>
> Specifically, these automatic metrics evaluate two crucial aspects of the generated pairs: the **instruction-following ability** and the **identity preservation** between the source and target images. We have taken rigorous steps to enhance data quality by **filtering out** **approximately 95%** of the generated data pairs that did not meet our metric's criteria.
>
> For further validation, we conducted a **human evaluation on a subset of the dataset** (200 pairs) to demonstrate the **alignment between our automatic metrics and human evaluators' assessments.** The results show a high degree of concordance, which underscores the reliability of our automatic filtering process. Below is a summary of the evaluation:
>
> |  | Success Cases | Failed Cases | Metric filtered |
> | --- | --- | --- | --- |
> | With Safeguard | 188 | 12 | \ |
> | Without Safeguard | 16 | 184 | 192 |
>
> We believe that these measures robustly address the concerns raised and further illustrate the efficacy and flexibility of our methodology.
>
> We trust that this additional information addresses your concerns, and we welcome any further inquiries or feedback you may have.
>
> ### **Q2: automatically generated region-based editing data may still occasionally exhibit inaccuracies or inconsistencies; maynot generalize across all editing scenarios**
>
> > Additionally, despite the team's efforts to provide high-quality region annotations, the automatically generated region-based editing data may still occasionally **exhibit inaccuracies or inconsistencies.** Furthermore, although experiments demonstrate the advantages of using real image anchors and region-based editing data, the improvements shown by the diffusion-based editing baselines are **specific to these benchmarks and may not generalize across all editing scenarios**. Future work should focus on enhancing the precision of region annotations and validating the dataset's applicability across a broader range of editing tasks.
> >
>
> ### A2：
>
> To mitigate the noise introduced by **inaccuracies** in Grounding DINO and SAM masks, we employ a comprehensive filtering and correction process:
>
> 1. **Identification of Inaccurate Masks**: We observe that **inaccurate masks** typically fall into the **following categories** (Examples in **the Figure 17 of the general response PDF**):
>     1. Incorrect identification leads to **overly large masks**.
>     2. Masks are **too small** to enable effective editing.
>     3. **Fragmented masks** due to segmentation failures.
>     4. **Fine-grained segment** masks too similar to the original object, complicating the editing process.
> 2. **Filtering and Adjustment:**
>     1. We **filter out** overly large, small, and fragmented masks. And the fine-grained masks are adjusted **using a soft mask,** which may be either a bounding box mask or a contour mask, to enhance the editing process and make the transformation of the mask boundary smooth.
> 3. Quality Assurance: Additionally, we employ **automatic metrics** to evaluate the final generated data's quality. This **end-to-end evaluation** process ensures the dataset's reliability, minimizing the influence of inaccuracies from Grounding DINO and SAM.
>
> To ensure the **generalization ability** of our dataset, we incorporate **human-written instructions** as in-context learning examples, addressing the limitations observed in large language models. Our process involves:
>
> 1. We first **manually created** a few hundred editing instructions to set a robust baseline. Then, we use these instructions to guide the language model in **generating additional instructions**, followed by a **meticulous manual review** to filter out subpar outputs. The process results in a curated set of **10k high-quality instruction pairs**, extensively detailed in **Table 2** of the **supplemental material.**
> 2. These **10k manually reviewed instruction pairs** are then used as **in-context learning examples** to further drive the language model to **producing instructions with substantial diversity**, ensuring coverage across a wide range of tasks compare to other datasets, as shown in **Table 3.**
>
> The **further experimenta**l results underscore the **instruction diversity and quality of our pipeline**.  The model trained with our datasets shows **superior performance** on the **Emu Edit Benchmark**, which includes **2000 cases across 8 distinct tasks** and we can see the model **showing superior performance in each task.**
> Thanks for **posing this valid question** and we will include **further details** on enhancing the precision of region annotations and validating the dataset's applicability across different tasks in the camera-ready version of the paper.
>
> We appreciate your thoughtful consideration of this aspect, and your feedback continues to enrich our understanding and future research directions. We also hope that you can kindly increase your score if our response has helped address your concerns.

---

> ### Author Response · Authors · 2024-08-20
>
> Dear Reviewer RFxS,
>
> We sincerely appreciate your careful review and valuable feedback, which has substantially improved our paper.
>
> Your feedback is highly valuable to me, and I want to make sure my responses effectively address the issues you have had. If you have any further questions or require additional clarification, please feel free to reach out.
>
> We also hope that you can kindly increase your score if our response has helped address your concerns.
>
> Best,
>
> Paper991 Authors

---

> ### Author Response · Authors · 2024-08-26
>
> Dear Reviewer RFxS,
>
> **We sincerely thank you for your insightful and constructive feedback on our manuscript.** Your comments have been invaluable in improving our work's clarity, depth, and overall quality. We have carefully considered each point and made substantial revisions to address your concerns and suggestions.
>
> We are eager to hear your thoughts on whether our revisions have effectively addressed your concerns. If you have any questions, feel free to discuss them.
>
> Once again, thank you for your valuable suggestions and the time you have dedicated to this process. We look forward to receiving your feedback.
>
> Kind regards, The authors

---

### Official Review · Reviewer_xmAL · 2024-08-01
**The Review of UltraEdit**

**Rating:** 6
**Confidence:** 4
**Correctness:** Yes.
**Clarity:** Yes.

**Review:**

The editing pipeline reasonably utilizes language models, segmentation models, and text-to-image models to create UltraEdit dataset, which ends up to a large-scale dataset with 4M editing samples with 750K instructions across 9+ editing types.

However, deep learning model-generated datasets may contain many artifacts due to the models’ hallucination or insufficient capability, especially when plenty of models are used to construct a single pipeline. Thus, I recommend a more detailed pipeline design and the incorporation of error prevention mechanisms. Details are in “Opportunities For Improvement.”

**Strengths:**

- The integration of language models, segmentation models, and text-to-image models is reasonable for building a strong instruction-based editing pipeline.
- The paper builds a large-scale dataset for instruction-based editing.
- The paper is easy to read.
- Results for models trained on ULTRAEDIT present promising results.

**Additional Feedback:**

See Opportunities For Improvement*.

**Documentation:**

Yes

**Limitations:**

Yes, the authors addressed the limitations and potential negative societal impact.

**Opportunities For Improvement:**

- The pipeline has some possible improvement directions:
    - In stage II’s data collection, datasets such as COCO and NoCaps contain mainly daily life images, which may not completely suitable for image editing tasks. A larger range of data source can help with the construction of a dataset with a rich distribution.
    - In stage II, the model used for editing, P2P, is only suitable for texture editing. Using different models in different editing types can lead to a better result. See PnP Inversion[1], InfEdit[2], MasaCtrl[3], and  Pix2Pix-Zero[4].

        [1] Ju X, Zeng A, Bian Y, et al. Pnp inversion: Boosting diffusion-based editing with 3 lines of code[C]//The Twelfth International Conference on Learning Representations. 2024.

        [2] Xu S, Huang Y, Pan J, et al. Inversion-free image editing with natural language[J]. CVPR 2024

        [3] Cao M, Wang X, Qi Z, et al. Masactrl: Tuning-free mutual self-attention control for consistent image synthesis and editing[C]//Proceedings of the IEEE/CVF International Conference on Computer Vision. 2023: 22560-22570.

        [4] Parmar G, Kumar Singh K, Zhang R, et al. Zero-shot image-to-image translation[C]//ACM SIGGRAPH 2023 Conference Proceedings. 2023: 1-11.

    - In stage II, why is using the diffusion pipeline with the noise-perturbed latent embedding important? For inversion, many techniques, such as null-text inversion[5], negative-prompt inversion[6], and PnP Inversion[1], can generate better results.

        [5] Mokady R, Hertz A, Aberman K, et al. Null-text inversion for editing real images using guided diffusion models[C]//Proceedings of the IEEE/CVF Conference on Computer Vision and Pattern Recognition. 2023: 6038-6047.

        [6] Miyake D, Iohara A, Saito Y, et al. Negative-prompt inversion: Fast image inversion for editing with text-guided diffusion models[J]. arXiv preprint arXiv:2305.16807, 2023.

    - In stage III, region-based editing could include a bounding box for object addition (editing type with no specific region).
    - In stage III, there are better inpainting pipeline options such as BrushNet[7], PowerPaint[8]. Moreover, the modified inpainting pipeline (algorithm 1) may disturb diffusion distribution. Can you provide some evidence for doing so?

        [7] Ju X, Liu X, Wang X, et al. Brushnet: A plug-and-play image inpainting model with decomposed dual-branch diffusion[J]. arXiv preprint arXiv:2403.06976, 2024.

        [8] Zhuang J, Zeng Y, Liu W, et al. A task is worth one word: Learning with task prompts for high-quality versatile image inpainting[J]. arXiv preprint arXiv:2312.03594, 2023.

    - Although a little later than the dataset, an instruction-based image editing solution[9] in the CVPR workshop may provide some reference for the editing pipeline (e.g., a separate pipeline for different editing types).

        [9] Ju X, Zhuang J, Zhang Z, et al. Image Inpainting Models are Effective Tools for Instruction-guided Image Editing[J]. arXiv preprint arXiv:2407.13139, 2024.

- A detailed user study or human-in-the-loop quality checking may better help users understand the overall quality of the dataset.

**Relation To Prior Work:**

There is some prior work missing. See Opportunities For Improvement*.

**Summary And Contributions:**

This paper presents a 4M generated dataset for instruction-based image editing named UltraEdit. UltraEdit has several distinct advantages over previous works: (1) a broader range of editing instruction, (2) better data sources, and (3) providing region-based annotations. Specifically, UltraEdit uses LLMs along with in-context human-written instruction examples to build editing instructions and target image prompts. And use prompt-to-prompt with text-to-image diffusion models to produce edited images. Moreover, it uses LLMs and Grounded SAM to produce an editing region for guidance in editing. The model trained on UltraEdit achieves new records on existing editing benchmarks.

Contributions:

- This paper proposes a novel pipeline for instruction-based image editing
- This paper builds a large-scale image editing dataset with diverse editing instructions and editing region annotations

---

> ### Author Rebuttal · Authors · 2024-08-17
>
> ### **Q1: Possible many artifacts due to the models’ hallucination or insufficient capability, how to prevent it.**
>
> > However, deep learning model-generated datasets may contain many artifacts due to the models’ hallucination or insufficient capability, especially when plenty of models are used to construct a single pipeline.  Thus, I recommend a more detailed pipeline design and the incorporation of **error prevention mechanisms**.
> >
>
> ### **A1:**
>
> Thank you for noticing the error prevention mechanisms in our method. As **elucidated in Section 2.3** of our manuscript, we employ an **automatic metric for data filtering**, which acts as a **safeguard** during data generation.
>
> Specifically, these automatic metrics evaluate **two crucial aspects** of the generated pairs: the **instruction-following ability** and the **identity preservation** between the source and target images. We have taken rigorous steps to enhance data quality by **filtering out approximately 95%** of the generated data pairs that did not meet our metric's criteria.
>
> For further validation, we conducted a **human evaluation** on a subset of the dataset (200 pairs) to demonstrate the **alignment between our automatic metrics and human evaluators**' assessments. The results show a high degree of concordance, which underscores the reliability of our automatic filtering process. Below is a summary of the evaluation:
>
> |  | Success Cases | Failed Cases | Metric filtered |
> | --- | --- | --- | --- |
> | With Safeguard | 188 | 12 | \ |
> | Without Safeguard | 16 | 184 | 8 |
>
> We believe that these measures robustly address the concerns raised and further illustrate the efficacy and flexibility of our methodology.
>
> ### **Q2. Limited Image source.**
>
> > In stage II’s data collection, datasets such as **COCO and NoCaps contain mainly daily life** **images**, which may not completely suitable for image editing tasks. A larger range of data source can help with the construction of a dataset with a rich distribution.
> >
>
> ### **A2:**
>
> To ensure diversity in our image dataset, we have sourced real images from a variety of datasets **beyond COCO and NoCaps**. As detailed in **Table 1 (supplemental material)**, besides the coco and nocaps, our real image sources include a diverse range: **Flickr, VizWiz, TextCaps, LN, ShareGPT4V** (which aggregates images from **LAION, SAM, GQA, OCR-VQA, TextVQA, VG, and web sources**), and **LAION**.
>
> We fully recognize the importance of image diversity for robust dataset construction. The modular and adaptable nature of our **pipeline** allows for the **continuous integration of diverse image sources**. This flexibility ensures that we can further enhance the dataset’s richness and applicability to a wide range of image editing tasks.
>
> We addresses the concern about source limitation by **incorporating extensive and varied datasets**, and we remain **committed to expanding this diversity** in future iterations.
>
> ### **Q3: Using different models in different editing types can lead to a better result**
>
> > In stage II, the model used for editing, P2P, is only suitable for texture editing. Using different models in different editing types can lead to a better result. See PnP Inversion[1], InfEdit[2], MasaCtrl[3], and Pix2Pix-Zero[4].
> >
> >
> > [1] Ju X, Zeng A, Bian Y, et al. Pnp inversion: Boosting diffusion-based editing with 3 lines of code[C]//The Twelfth International Conference on Learning Representations. 2024.
> >
> > [2] Xu S, Huang Y, Pan J, et al. Inversion-free image editing with natural language[J]. CVPR 2024
> >
> > [3] Cao M, Wang X, Qi Z, et al. Masactrl: Tuning-free mutual self-attention control for consistent image synthesis and editing[C]//Proceedings of the IEEE/CVF International Conference on Computer Vision. 2023: 22560-22570.
> >
> > [4] Parmar G, Kumar Singh K, Zhang R, et al. Zero-shot image-to-image translation[C]//ACM SIGGRAPH 2023 Conference Proceedings. 2023: 1-11.
> >
>
> ### **A3:**
>
> Thank you for pointing out **these insightful works**. We will **ensure they are included in our paper**. We appreciate the suggestion to explore a broader range of models.
>
> Our UltraEdit pipeline is inherently **modular and adaptable**, allowing for **integration of various image editing models**.
>
> In our current framework, our method is **particularly effective**; leveraging SDXL-turbo, each sample is generated in just **1-4 steps**, which is **significantly faster** than previous methods **by magnitudes**. To be notice that this efficiency **does not compromise the quality**—the inclusion of **real image anchors** in our dataset ensures **robust identity preservation and high-quality results**, as shown in **Figure 8 of the Supplemental material** .
>
> We have i**ndeed tested the all the mentioned methods** and we will **incorporate them into our pipeline**.  Experiment results are provided in the **Figure 15 of our general response PDF**, confirming that our method **remains highly competitive**. We are committed to **presenting a detailed comparison** in the camera-ready version of our paper. Thank you again for your constructive feedback.

---

> > ### Author Rebuttal · Authors · 2024-08-17
> >
> > ## **Rebuttal(2/n) by Authors**
> >
> > ### **Q4: Why use image as anchor rather than using the null-text inversion?**
> >
> > > In stage II, why is using the diffusion pipeline with the noise-perturbed latent embedding important? For inversion, many techniques, such as null-text inversion[5], negative-prompt inversion[6], and PnP Inversion[1], can generate better results.
> > >
> > >
> > > [5] Mokady R, Hertz A, Aberman K, et al. Null-text inversion for editing real images using guided diffusion models[C]//Proceedings of the IEEE/CVF Conference on Computer Vision and Pattern Recognition. 2023: 6038-6047.
> > >
> > > [6] Miyake D, Iohara A, Saito Y, et al. Negative-prompt inversion: Fast image inversion for editing with text-guided diffusion models[J]. arXiv preprint arXiv:2305.16807, 2023.
> > >
> >
> > ### **A4:**
> >
> > Thank you for the **insightful references**; we'll **include a discussion of these works** in the final version.
> >
> > Our approach to using real image anchors to guide image generation rather than using the null-text inversion ****serves two primary purposes:
> >
> > 1. **Avoidance of Bias from image sources**: Although we gathered real images from diverse sources, using **real images directly could introduce inherent biases** present in the original image dataset, **limiting the generalizability** of our model. For example, only suitable for real image editing and weak in others. By using real images as anchors, we **ensure a diverse and controlled** dataset, **minimizing image biases** of the generated dataset.
> > 2. **Efficiency**: Editing real images typically involves **intensive processes like null-text inversion**, which requires **hundreds of steps per image**. Our method, on the other hand, can generate high-quality image editing pairs i**n just 1 to 4 steps** **using the real image as anchors**. This efficiency makes our approach more **scalable and cost-effective**, facilitating the generation of a **large corpus of editing samples without compromising on quality or fidelity**.
> >
> > Our methodology not only ensures a **more unbiased and diverse** dataset but also **streamlines the editing process**, making it **significantly more efficient without compromising on image fidelity**. We aim to generate **efficient and cost-free methods** for data generation and **more data leads to more powerful results.**
> >
> > We also tested **all the mentioned methods** and we will make sure to **thoroughly reference these highly related works**.  Experiment results in **Figure 15 of our general response PDF** confirm that our method performs **competitively with high efficiency.**
> >
> > ### **Q5:**
> >
> > - **In stage III, region-based editing could include a bounding box for object addition (editing type with no specific region)**
> >
> > ### **A5:**
> >
> > Thank you for highlighting this aspect. Our pipeline does support object addition in region-based editing, as illustrated in Figure 6. We will include more examples and implementation details for object addition of our data generation pipeline in the final version to enhance the clarity and comprehensiveness of our paper.
> >
> > ### **Q6: In stage III, there are better inpainting pipeline options such as BrushNet[7], PowerPaint[8]. Moreover, the modified inpainting pipeline (algorithm 1) may disturb diffusion distribution. Can you provide some evidence for doing so?**
> >
> > > [7] Ju X, Liu X, Wang X, et al. Brushnet: A plug-and-play image inpainting model with decomposed dual-branch diffusion[J]. arXiv preprint arXiv:2403.06976, 2024.
> > >
> > >
> > > [8] Zhuang J, Zeng Y, Liu W, et al. A task is worth one word: Learning with task prompts for high-quality versatile image inpainting[J]. arXiv preprint arXiv:2312.03594, 2023.
> > >
> >
> > ### **A6:**
> >
> > Thank you for pointing out **these insightful works**. We will ensure they are **included in our paper**. We appreciate the opportunity to consider these methods for **potential integration** into our pipeline.
> >
> > In Stage III, we use our proposed method (Algorithm 1) for generating region-based images to **ensure a smooth transition** between the inpainting area and the rest of the image.  It is the same reason we use a **soft mask** for image generation. During our data circulation, we notice that the our methods can significantly **mitigate artifacts and abrupt boundaries** between the mask region and the the rest of the image. Qualitative evaluations presented in **Figure 16 of the general response PDF** show the **effectiveness** of our proposed method.
> >
> > Furthermore, we have explored **all the aforementioned methods** for image generation and and **plan to integrate them into our pipeline**. We also provided the experiment results of these methods in **Figure 15 of the general response PDF**, confirming that our **methods remains highly competitive**.
> >
> > We are committed to **including a detailed comparison and human evaluation** in the camera-ready version of our paper. Thank you again for your insightful feedback and suggestions.
> >
> > ### **Q7:**
> >
> > - **Although a little later than the dataset, an instruction-based image editing solution[9] in the CVPR workshop may provide some reference for the editing pipeline (e.g., a separate pipeline for different editing types).**
> >
> >     > [9] Ju X, Zhuang J, Zhang Z, et al. Image Inpainting Models are Effective Tools for Instruction-guided Image Editing[J]. arXiv preprint arXiv:2407.13139, 2024.
> >     >
> >
> > ### **A7:**
> >
> > Thank you for bringing this work to our attention. It provides **valuable insights** into instruction-based image editing, particularly regarding editing pipeline strategies. We will **ensure we cite this work and consider its approaches** to refining our methods.

---

> > ### Author Rebuttal · Authors · 2024-08-17
> >
> > ## Rebuttal(3/n) by Authors
> >
> > ### **Q8: User study and human-evalutaion?**
> >
> > > A detailed user study or human-in-the-loop quality checking may better help users understand the overall quality of the dataset.
> > >
> >
> > ### **A8:**
> > Thank you for emphasizing the importance of user studies and human evaluation.
> >
> > As detailed in the **Supplemental material**, we **have conducted comprehensive human evaluations** on models trained with our dataset. Human evaluation shows in **Table 7 in the Supplemental material** showcases the superior quality of UltraEdit compared to other dataset.
> >
> > |  | Ours | MagicBrush | InstructPix2Pix |
> > | --- | --- | --- | --- |
> > | TrueSkill score | 25.5 ± 0.8 | 23.7 ± 0.7 | 22.6 ± 0.7 |
> >
> > We also evaluate the **performance of various model backbones** trained on our dataset using human evaluation.
> >
> > |  | SD3 w/ UltraEdit | SDXL w/ UltraEdit | SD1.5 w/ UltraEdit | Emu Edit |
> > | --- | --- | --- | --- | --- |
> > | TrueSkill score | 26.7 ± 0.7 | 26.5 ± 0.7 | 26.0 ± 0.7 | 25.1 ± 0.7 |
> >
> > We conducted a **human evaluation on a subset of the dataset** (200 pairs) to demonstrate the **quality** of our dataset. Below is a summary of the evaluation:
> >
> > |  | Success Cases | Failed Cases | Metric filtered |
> > | --- | --- | --- | --- |
> > | With Safeguard | 188 | 12 | \ |
> > | Without Safeguard | 16 | 184 | 192 |
> >
> > These evaluations underscore the effectiveness and high quality of UltraEdit, ensuring that it meets the real-world needs of users. We will continue to **refine our evaluation processes** and include more detailed analyses in the camera-ready version of our paper. Thank you for your valuable suggestion.
> >
> >
> > We hope this detailed explanation successfully addresses your concern. Your advice and scrutiny have significantly contributed to the quality of our paper, and we are sincerely grateful for your time and expertise.

---

> ### Author Response · Authors · 2024-08-20
>
> Dear Reviewer xmAL,
>
> We sincerely appreciate your careful review and valuable feedback, which has substantially improved our paper.
>
> Your feedback is highly valuable to me, and I want to make sure my responses effectively address the issues you have had. If you have any further questions or require additional clarification, please feel free to reach out.
>
> We also hope that you can kindly increase your score if our response has helped address your concerns.
>
> Best,
>
> Paper991 Authors

---

> ### Author Response · Authors · 2024-08-22
>
> Dear Reviewer xmAL,
>
> Your feedback is highly valuable to me, and I want to make sure my responses effectively address the issues you have had. If you have any further questions or require additional clarification, please feel free to reach out. We also hope that you can kindly increase your score if our response has helped address your concerns.
>
> Best,
>
> Paper991 Authors

---

> ### Author Response · Authors · 2024-08-26
>
> Dear Reviewer xmAL,
>
> We appreciate your constructive feedback and observations! Kindly inform us if you find our response acceptable, and we'll be eager to address any additional concerns you might still harbor.
>
> Sincerely,
>
> Authors.

---

> ### Author Response · Authors · 2024-08-31
>
> Dear Reviewer xmAL,
>
> Thank you for your thoughtful review of our manuscript. We have carefully addressed your comments in our rebuttal and made revisions to improve the work.
>
> As the discussion period ends in 1 days, we kindly request that you review our rebuttals to see if they address your concerns or if you have any further insights. Your expertise is greatly appreciated.
>
> Sincerely,
>
> Authors of Paper991

---

### Author Rebuttal · Authors · 2024-08-17

## General Response：


Dear Reviewers,

We thank all reviewers for their insightful comments and acknowledgment of our contributions.

We greatly appreciate your recognition of the strengths of our work as follows:

### **Novel Dataset and Generation Pipeline**

We present UltraEdit, recognized as a "**novel**" and "**meaningful**" dataset for instruction-based image editing by `j7cR` and `3R87` , containing "**diverse** editing instructions and editing region annotations" (`xmAL`). Showing "**clear motivation**" (`fz2t`), it is a large-scale, automatically generated dataset that "**far exceeds** prior datasets" and "**addresses the drawbacks** in existing image editing datasets" (`RFxSL`). The data generation pipeline has also been recognized by **most reviewers**(`xmAL`, `RFxSL`, `j7cR`) as a "**novel**" approach .

### **Comprehensive Experiments and Analysis:**

Our work is acknowledged for its **comprehensiveness** and **promising results** by **all reviewers**. The "comprehensive comparisons and ablations" (`3R87`) offer "**clear insights**" (`RFxSL`) and "sets **new records**" (`RFxSL`, `xmAL`) in this field.

### **Contribution to the Community:**

**All reviewers** hold positive views towards this work, acknowledging our "**significant**" and "**beneficial**" **contribution** to the community (`RFxSL`, `3R87`, `j7cR`). It serves as a "better starting point" (`3R87`) and a "valuable resource" (`RFxSL`) for future research. Our dataset is highlighted by `fz2t` as open-source and well-documented.

Overall, we are pleased to note that **all reviewers** hold positive opinions towards this work and all agree that the work is **above the acceptance threshold**.

We have made detailed responses to each reviewer's concerns, which are carefully addressed point-by-point.

#### **Summary of General Responses:**

- (`xmAL` , `3R87`, `RFxS`)**Error Prevention Mechanisms and Safeguards:**
    As elucidated in Section 2.3 of our manuscript, we employ an **automatic metric** for data filtering, which acts as a **safeguard** during data generation, filtering out approximately 95% of the generated data pairs. We further conducted a **human evaluation** on a subset of the dataset (200 pairs) to demonstrate the alignment between our automatic metrics and human evaluators' assessments.

|  | Success Cases | Failed Cases | Metric filtered |
| --- | --- | --- | --- |
| With Safeguard | 188 | 12 | \ |
| Without Safeguard | 16 | 184 | 192 |

- (`j7cR`, `fz2t`)**Bias in LLM-Generated Editing Instructions:**

    As detailed in Section 2 and Appendix A2, we incorporate **human-written instructions as in-context learning examples** to address limitations. Our pipeline doesn't solely rely on LLMs but combines human intelligence and machine efficiency to mitigate bias. The model trained with our datasets shows superior performance on the Emu Edit Benchmark across 8 tasks, demonstrating the generalization ability of our dataset.

- (`3R87`)**Fair Comparison with Our Datasets:**

    As clarified in line 145 in Section 3 of our paper, we use the **same diffusion model architecture** (Stable Diffusion v1.5) as the backbone for all experiments to maintain consistency with other baselines. We also employ **training data volumes identical** to those used in other methods.

- (`j7cR`, `3R87`) **Evaluations:**

    Following previous works, we use **CLIP and DINO similarity** measures to ensure **similarity** between the source and target images, while using **CLIP-TEXT similarity and CLIP-direction** to evaluate the **changes between images**.

- (`xmAL`) **Comparison with Other Methods for Data Generation:**

    We have made an comparison with **all the methods** suggested by Reviewer `xmAL` with our pipeline, with results shown in **Figure 15** of the **attached PDF.** We are committed to incorporating these useful methods, namely: **Null-text inversion; Pnp inversion; Brushnet; Powerpaint; MasaCtrl; InfEdit**, into our pipeline and thoroughly referencing these highly related works.

- (`xmAL`, `fz2t`) **Reasons for Not Using Real Images for Building the Dataset:**

    This approach serves two primary purposes:
    - **Avoidance of Bias from Image Sources**
    - **Efficiency Concerns:** Our pipeline generates high-quality image editing pairs requiring significantly fewer steps (1-4 steps) using real images as anchors. Experiment results in Figure 15 of response PDF confirm that our method performs significantly efficient without compromising on image fidelity.


- (`xmAL`, `RFxS`, `fz2t`, `3R87`) **Clarification of the Region-Based Image Editing Pipeline:**

    To mitigate noise introduced by extra segment models in the pipeline, we employ a thorough filtering and correction process. We observed **inaccurate masks** typically falling into four categories (illustrated in **Figure 17 of the attached PDF**), and we made the corresponding  filtering and adjustments.

    Additionally, we provide an **ablation study** of the region-based image editing pipeline in **Figure 16 of the attached PDF**. Generated images without our method exhibit noticeable artifacts along the boundaries of the original and edited regions, emphasizing pronounced border effects.

We believe our work could make a novel contribution to the community and offer insights on key design principles.

We are eager to engage in further discussions if any additional questions are raised.

Best regards,

Authors.

---

### Decision · Program_Chairs · 2024-09-26

**Decision:**

Accept (Poster)

**Comment:**

### Summary
This paper introduces UltraEdit, a large-scale dataset automatically generated for instruction-based image editing. UltraEdit handles various fine-grained editing instructions and offers a new type of instruction: region-based instruction. The model trained on UltraEdit achieves good performance on image editing benchmarks.

### Strengths
1. The paper proposes a scalable dataset pipeline and the authors release the large-scale instruction-based editing data.
2. The paper expands the boundaries of image editing, demonstrating actual performance improvements.
3. The use of region-based annotations enables an interesting and practical use-case, localized editing.

### Weaknesses
1. Using LLMs and generative methods (e.g., Stable Diffusion) to create synthetic data for improving performance on specific tasks is not a new paradigm, thus, it does not provide significantly new insights.
2. The proposed framework's components can be seen as a well-combined set of existing modules rather than novel contributions.
3. There are concerns about the potential side effects of using generated images for training.

### Meta-Review
This paper conducted active discussions between the authors and reviewers during the rebuttal phase. Overall, reviewers expressed positive opinions about the work. The concerns about the generated images for training were adequately addressed during the rebuttal.
Based on the observations, reviews, and discussions, the Area Chair has decided to accept this paper. This work presents a valuable contribution to the field of instruction-based image editing, offering a new large-scale dataset and demonstrating its effectiveness in improving model performance.

However, the paper would be improved by resolving and reflecting on the issues raised by reviewers to gain further impact. Specifically, the authors are encouraged to elaborate on the novelty of their approach, analyze the implications of using generated images for training, and discuss the broader impact of their work on image editing and multimodal AI.